# TWINNED INTERVENTIONAL FLOWS

## ABSTRACT

Real-world problems in continuously evolving settings, such as predicting the efficacy of medical treatment, often require estimating the causal effects of interventions. Issues such as irregularly-sampled and missing data, unobserved factors, and ethical concerns make such settings especially challenging. The existing methodology relies on low-dimensional embeddings, potentially incurring information loss.

We circumvent this limitation with a novel approach "twinning" that augments the partial observations with additional latent variables and appeals to conditional continuous normalizing flows to model the system dynamics, obtaining accurate density estimates. We also introduce a new approach to overcome a key technical challenge, namely, mitigating stiffness of the underlying neural ODE. The model provably benefits from auxiliary non-interventional data during training. We showcase the flexibility of the proposed method with tasks like anomaly detection and counterfactual prediction, and benchmark on standard reinforcement learning (Half-Cheetah) and treatment effect prediction (tumor growth) contexts.

## 1 INTRODUCTION

Adaptive decision making in continuously evolving systems remains a key challenge in several domains (e.g., healthcare) that require intermittent interventions. Continuous normalizing flows (CNFs) have emerged as methods of choice in such temporal settings since they can accommodate longitudinal data with observations arising at irregular intervals (Rezende & Mohamed, 2015; Chen et al., 2018; Morrill et al., 2021; Javaloy et al., 2023). Importantly, CNFs can be leveraged as effective generative models for extrapolation, i.e., to make predictions about future (e.g., medical condition of a patient). CNFs learn complex distributions from simpler ones (such as Gaussian) via a sequence of invertible transformations, so enable exact likelihood estimates. Such estimates are particularly appealing since they can be used, e.g., to rule out unlikely events and detect anomalies.

However, typically, such settings entail some unobserved *confounders* (Pearl, 2009) that may result in undesirable distributional shifts; e.g., an unknown medical condition can interfere with an otherwise appropriate treatment (Bennett & Kallus, 2021). The prevalent paradigm is to embed data into a low-dimensional space, and learn a flow treating confounders as (time-varying) latent variables in this space (Seedat et al., 2022; De Brouwer et al., 2022). Low-dimensional inference, however, impedes the ability of these models to provide accurate density estimates even with powerful CNFs.

Several other issues, such as data sparsity, often compound the problem: collecting the data might be expensive, risky, or raise ethical concerns. For instance, medical trials seek to find an effective treatment but conducting many explorative experiments are not ethically acceptable as they might put a patient at risk. In such scenarios, some auxiliary data (e.g., the medical history of the patient prior to their engagement in the trials) can help mitigate the need for interventional samples (Ilse et al., 2021). However, such passive *observational* data may contribute additional confounding effects and be misaligned with the interventional distribution, which can cause additional identifiability problems.

We address all these real-world challenges with a novel approach "twinning", that augments partial observations with additional latent variables including confounders as a state and treats interventions as actions in a Partially-Observable Markov Decision Process (POMDP) model. Specifically, we define a conditional flow, namely *Twinned Interventional Flow* (TIF), over this state space to model the system dynamics.

Table 1: **A summary of capabilities of TIF (proposed in this work), and other methods**. TIFs can perform several tasks, e.g., estimate log densities (*logp*), learn causal effects of interventions (*causal ef.*), include unobserved confounders as part of the model design (*conf.*), leverage additional observational data (*obs. data*), accommodate irregular and missing observations (*irreg. samp.*), and make online predictions (*online pred.*).

| Method | logp | causal ef. | conf. | obs. data | irreg. samp. | online pred. |
|---|---|---|---|---|---|---|
| Du et al. (2020) | ✗ | ✗ | ✓ | ✗ | ✓ | ✓ |
| Yildiz et al. (2021) | ✓ | ✗ | ✗ | ✗ | ✓ | ✓ |
| Ilse et al. (2021) | ✗ | ✓ | ✓ | ✓ | ✓ | ✗ |
| Bennett & Kallus (2021) | ✗ | ✓ | ✓ | ✓ | ✓ | ✓ |
| Khemakhem et al. (2021) | ✓ | ✓ | ✗ | ✗ | ✗ | ✓ |
| Zhu et al. (2022) | ✗ | ✓ | ✓ | ✓ | ✗ | ✓ |
| Seedat et al. (2022) | ✗ | ✓ | ✓ | ✗ | ✓ | ✓ |
| De Brouwer et al. (2022) | ✗ | ✓ | ✓ | ✗ | ✓ | ✓ |
| **TIF** (this work) | ✓ | ✓ | ✓ | ✓ | ✓ | ✓ |

While developing TIFs, we faced challenges with stiffness in the ODE. Stiff equations, as defined by Hairer & Wanner (1996), are problematic for which explicit methods don't work. Stiff ODEs pose numerical issues in inversion, especially with the adjoint method, where the error in the reverse pass using explicit methods can be significantly larger than in the forward pass (Kim et al., 2021; Gholaminejad et al., 2019). Stiffness can jeopardize log-likelihood estimation if the initial distribution is not recovered. We address this by proposing a novel penalty term to mitigate stiffness in the ODE in this paper.

We establish a solid theoretical foundation for accurate log-likelihood estimation using TIF. TIFs also provide *provable* benefits with observational data: observational data, *even* when missing some features, can be leveraged to obtain a strictly better estimator, asymptotically, for POMDPs than what is achievable by limited interventional data alone. Table 1 underscores the flexibility of TIF. For instance, in a medical setting, TIF can learn causal effects of interventions (e.g., treatments), account for unobserved confounders that affect the system (e.g., the background and experience of doctor, comorbidities), handle irregular and missing observations (e.g., missing or delayed X-ray reports) effectively, and perform online predictions (e.g., about the health of a patient). Additionally, TIF is well-equipped to handle causal inquiries, such as counterfactual prediction.

To substantiate the conceptual and theoretical merits of TIF, we conducted a series of experiments across tasks such as anomaly detection and counterfactual prediction, and benchmark on standard reinforcement learning (Half-Cheetah) and treatment effect prediction (tumor growth) settings. TIF consistently demonstrated strong performance in these experiments.

## 1.1 OUR CONTRIBUTIONS

We now summarize the main contributions of this work. Here, we

- **(Conceptual)** propose a new technique *twinning* to enable, simultaneously, density estimates and causal queries in the presence of unobserved confounders;
- **(Methodological)** introduce Twinned Interventional Flows (TIFs) that bring together and find use in reinforcement learning, causal reasoning, generative models and neural ODEs (Table 1);
- **(Theoretical)** establish the benefit of using observational data for model learning in fully continuous POMDP settings (previously, such results known only for discrete settings) even when some observables are missing or masked;
- **(Technical)** emphasize the often overlooked problem of the stiffness of ODEs in the context of online prediction, and introduce a new penalty method with demonstrable benefits; and
- **(Empirical)** substantiate the efficacy of the proposed framework using detailed empirical investigations, including, on standard reinforcement learning and causal benchmarks.

We now proceed to reviewing relevant related works.

## 1.2 RELATED WORK

**Generative models and Neural ODEs** Our work is inspired by, and builds on, the the success of flow-based generative models (Abdal et al., 2021; Grathwohl et al., 2019; Yang et al., 2019; Dinh et al., 2015; Kingma & Dhariwal, 2018; Rezende & Mohamed, 2015; Verma et al., 2023; Papamakarios et al., 2021). CNFs do not require sophisticated regularization like the step-wise methods, since the neural ODE can be inverted by running the numerical solver backwards in time, and the flow of probabilities can be integrated into this process (Chen et al., 2018). Recently, deep connections have been shown between score-based generative models and diffusion models. In particular, the associated SDE can be expressed as an equivalent neural ODE thereby enabling exact likelihood computation (Song et al., 2021).

**Causal inference** Continuous-time dynamics in the context of causal inference (Pearl, 2009) have attracted attention. Seedat et al. (2022) model time-depending confounders, building on the work of Kidger et al. (2020), who use neural controlled differential equations for longitudinal data with observations. However, using natural cubic splines to approximate observations in a continuous space is not applicable for online prediction, so Morrill et al. (2021) advocate rectilinear control paths. These models embed into a lower dimensional space, so cannot always estimate accurately the densities of observations. De Brouwer et al. (2022) adapt Bayesian Neural Networks that maintain uncertainty over model parameters for causal treatment effect prediction, avoiding assumptions about the overlap between data sets with different treatments. Khemakhem et al. (2021) show the benefits of estimating log-densities and using flow-based models for a range of causal inference tasks. Their work focuses on regularly sampled data. Ilse et al. (2021) combine interventional and observational data to learn the causal effects, similar to this work. However, their method uses splines and so cannot enable online prediction, where stiffness of the system becomes a key obstacle.

**Reinforcement Learning (RL)** The continuous nature of many RL environments is well-established. Doya (2000) introduced a method to obtain an optimal policy in continuous-time dynamics. Yildiz et al. (2021) use Bayesian Neural ODEs to model continuous-time dynamics and evaluate uncertainties in the model. However, these approaches assume that the state is fully observed. Du et al. (2020) use Neural ODEs to model the latent space with CNF and solve RL problems in a continuous time POMDP but neither provide likelihood estimates nor exploit observational data. Interesting work has been done at the intersection of causal inference and RL, see e.g., Ding et al. (2022), Zhang & Bareinboim (2022), and Bareinboim et al. (2015). Zhu et al. (2022) study causal discovery with RL, and combine off-policy and on-policy data in discrete settings. Bennett & Kallus (2021) give extensive analyses and mathematical grounds for causal reasoning in the continuous setting. Hızlı et al. (2023) are also combining ideas from RL and causality by jointly modeling the treatment policy and the outcomes. We focus on motivating the benefits of flow models for POMDPs. The POMDP and the policy learning in such setting has been previously studied, e.g., from optimal filtering and optimal control (Alt et al., 2020) as well as belief state (Chen et al., 2022) perspectives.

## 2 PROBLEM FORMULATION

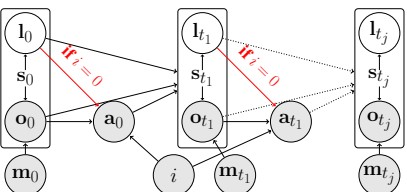

Figure 1: Underlying DAG of the POMDP. The red arrows indicate privileged information. Observed variables are marked with gray.

| Variable | Notation | Observed |
|---|---|---|
| State | $s_t \in \mathcal{S} \subset \mathbb{R}^n$ | ✗ |
| Obs. | $o_t \in \mathcal{O} \subset \mathbb{R}^m$ | at $t_j \in T$ |
| Latent | $l_t \in \mathcal{L} \subset \mathbb{R}^{n-m}$ | ✗ |
| Action | $a_t \in \mathcal{A} \subset \mathbb{R}^l$ | at $t_j \in T$ |
| Mask | $m_t \in \mathcal{M} \subset \mathbb{R}^m$ | at $t_j \in T$ |
| Indicator | $i \in \{0, 1\}$ | ✓ |

Table 2: Notation and details of the POMDP framework

Our primary objective is to develop a predictive model that accurately captures the system dynamics under various interventions and possible (unobserved) confounders. Identifying the number of

confounders or recovering a full causal graph is tedious and often infeasible. We therefore adopt an approach that does not constrain the causal hierarchy within observed and unobserved variables and can be described in terms of a continuous POMDP. We focus on studying a continuous dynamic system marked by irregular observations and interventions. Our dataset comprises both longitudinal interventional data ($\mathcal{D}_{int}$) and additional observational data ($\mathcal{D}_{obs}$).

This set of problems is motivated by real-world data scenarios and decision-making problems that necessitate leveraging imperfect data This is particularly relevant in domains such as medicine, where the need to make decisions using imperfect data is a common and critical aspect.

## 2.1 Partially Observable Markov Decision Process

We adopt a continuous proximal reinforcement learning paradigm that extends proximal causal inference to dynamic longitudinal settings while enabling identification in more general settings and providing efficient estimators. Specifically, we follow the approach of *two views of current observations* suggested by (Bennett & Kallus, 2021), where actions of the agent in observational data can be conditioned on auxiliary offline or *privileged* data, i.e., observations of the state that are not available for the intervening agent. As future observations or actions do not affect the current system dynamics we obtain a directed acyclic graph (DAG) (Pearl, 2009), illustrated in Figure 1.

We will use the standard *do*-notation (Pearl, 2009) as a means of distinguishing between observed and interventional actions. In particular, we use $do(a_t = a)$ to denote an external intervention that sets the action $a_t$ at time $t$ to the value $a$. As we assume that in $\mathcal{D}_{int}$, interventions depend solely on the observable variables, and any other connections to the node $a_t$ in the underlying Directed Acyclic Graph (DAG) (Pearl, 2009, p. 12-21) present in $\mathcal{D}_{obs}$ can be pruned. This pruned causal connection is denoted with red in Figure 1. Thus when mention interventional data we are referring to data generated from a policy based on observable variables, rather than a completely random policy. The difference in the causal graphs causes distributional shift between $\mathcal{D}_{int}$ and $\mathcal{D}_{obs}$.

We extend the POMDP $(\mathcal{S}, \mathcal{O}, \mathcal{A}, \mathcal{Z}, \mathcal{P})$ previously considered by Bennett & Kallus. Here transition probabilities $\mathcal{P}$ capture the dynamics of the system and the observation probabilities $\mathcal{Z}$ describe the noisy map between the true state and the observed values. The system state ($s_t$) evolves continuously in time $t \in \mathbb{R}^+$ as a joint distribution over the observables and latent variables including any confounders. Discrete-time observations ($o_{t_j}$), capture a subset of the state at irregular intervals $t_j \in T = (t_0, t_1, \ldots, t_L)$. To capture the remaining unobserved portion of the state space, we introduce a latent variable ($l_t$). Concurrently with the observations ($T$), we have access to the actions ($a_{t_j}$) applied to the system by the intervening agent. These actions exert a continuous effect over time, governed by an unknown function, $f(a_{t_j}, t - t_j)$, for $t \in [t_j, t_{j+1}]$.

We further introduce a masking variable ($m_{t_j}$) to indicate which observations are available at time $t_j$. We demonstrate that TIFs can cope with partially missing observations. Furthermore, we include an indicator $i$ to inform the model whether the data is observational ($i = 0$) or interventional ($do(a) \rightarrow i = 1$). Note that while the observations are made at discrete times $T$, the state ($s_t$) evolves continuously and the effect of action ($f(a_{t_j}, t - t_j)$) on the dynamics of the state is also continuous. We do not discuss the rewards of POMDP in this work as the policy optimization is not the focus of this study. All variables and whether they are observed are summarized in Table 2.

To refer to the previous action/observation time within the set $T$, we will use the notation $t_{-T}$, i.e., $t_{-T} = \max(t_j \in T; t_j < t)$. We also denote the history at time $t_j$ as $\mathbf{H}_{t_j} = (o_{t_0}, m_{t_0}, a_{t_0}, o_{t_1}, m_{t_1}, a_{t_1}, \ldots, m_{t_j}, o_{t_j})$. Namely, the history encompasses observations, masking indicators, and actions up to time $t_j$, but does not include $a_{t_j}$.

POMDP is closely tied with causal treatment effect prediction. Specifically, action $a_t$ serves the role of a treatment, and time-varying confounders can be included in $l_t$. Effectively predicting treatment effects implies that the problem can be cast as an RL task, where the objective is to optimize the treatment based on the predictions.

## 3 Twinned Interventional Flow (TIF)

In partially observable systems, applying CNF directly to observations (i.e., $s_t = o_t$) is insufficient to describe the full dynamics. Therefore, we "*twin*" the observations with a sufficient number of

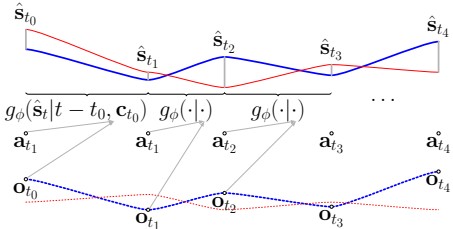
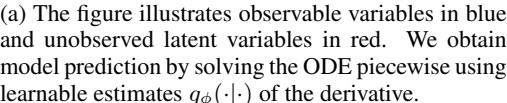
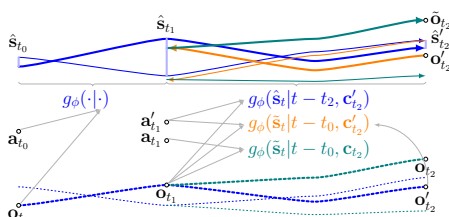

(a) The figure illustrates observable variables in blue and unobserved latent variables in red. We obtain model prediction by solving the ODE piecewise using learnable estimates $g_\phi(\cdot|\cdot)$ of the derivative.

(b) The figure showcases both the factual (blue) and counterfactual (green) trajectories of the system. The orange trajectory represents the backward run trajectory, initiated from the factual state.

Figure 2: (Left) Online prediction, and (right) Counterfactual prediction. The underlying dynamics are depicted at the bottom with dotted lines and the continuous prediction with solid lines at the top.

latent variables to construct an augmented observation and latent space $\hat{s}_t$ describing $s_t$. We call it as *twinned space*. Twinning allows us to obtain a bijective map for which conditioned CNF can be applied. In particular, we can model $\frac{\partial s}{\partial t}$ with a single neural network $g_\phi$ without additional encoding-decoding structure. Importantly, by avoiding encoding into a lower-dimensional space, the model remains invertible in time. Invertibility affords many benefits (Johansson et al., 2019). Particularly, in our context, it enables both counterfactual prediction illustrated in Figure 2b (details in the supplementary Section H) and accurate density estimates. Figure 2a visualizes the overall flow along with the continuous predictions generated by our model.

### 3.1 MODEL ARCHITECTURE

We model the effect of time $t - t_{-T}$ elapsed since previous action, and the context $c_{t_{-T}} := [a_{t_{-T}}, o_{t_{-T}}, m_{t_{-T}}, i]$ with a learnable model $g_\phi(\hat{s}_t|t - t_{-T}, c_{t_{-T}})$ , where $\hat{s}_t$ approximates the true state $s_t$. Consequentially, at time $t > t_j$ the approximated state becomes conditioned on $\mathbf{H}_{t_j}$ enabling an estimate of the true state. Our architecture inherits from recent works (Abdal et al., 2021; Grathwohl et al., 2019; Yang et al., 2019) that implement conditioned continuous normalizing flows with several gate-bias modulations (details in the supplementary materials (Section B).

Furthermore, $g_\phi$ can be used to obtain the model approximation of the change in the model state ($\hat{Q}$) from time $t_j$ to $t$ as,

$$\hat{Q}_{t_j, t} = \int_{\tau = t_{-T}}^{t} g_\phi(\hat{s}_\tau|\tau - t_j, c_{t_j})d\tau \quad, \forall t_j < t \leq t_{j+1} \,, \tag{1}$$

As the observations and actions are observed at discrete time points, we need to account for discontinuities at these points. Thus the flow needs to be divided into continuous sub flows that must be summed over to obtain the approximation of the state at time t. In particular, we get the state at time $t$ as $\hat{s}_t = \hat{s}_0 + \sum_{t_j \in T|t_j < t} \hat{Q}_{t_j, \min(t_{j+1}, t)}$ , where we define $t_{L+1} = \infty$ to account for time after the last observation $t_L$ in $T$. Hereon, we will not explicitly denote the sub-flows with the summation, as the numerical solvers can be informed about possible discontinuities of the derivative at the time of actions and observations.

### 3.2 THEORETICAL RESULTS

**Log-likelihood estimate**  Using the twinned space allows us to obtain a formula for the change in log densities under some assumptions that are standard in literature (described in the supplementary section C). We begin with a result that quantifies the effect of latent variables for our purpose.

**Theorem 3.1.** *The change in the log probabilities of observations in time can be expressed as an expected value of the trace over the latent variables $l_t$.*

$$\frac{\partial}{\partial t} \log(p(o_t|\mathbf{H}_{t_{-T}}, a_{t_{-T}}, i)) = \mathbb{E}_{p(l_t|o_t, \mathbf{H}_{t_{-T}}, a_{t_{-T}}, i)} \left[ -Tr\left( \frac{\partial}{\partial s} g_\phi(s_t, |t - t_{-T}, c_{t_{-T}}) \right) \right] \,.$$

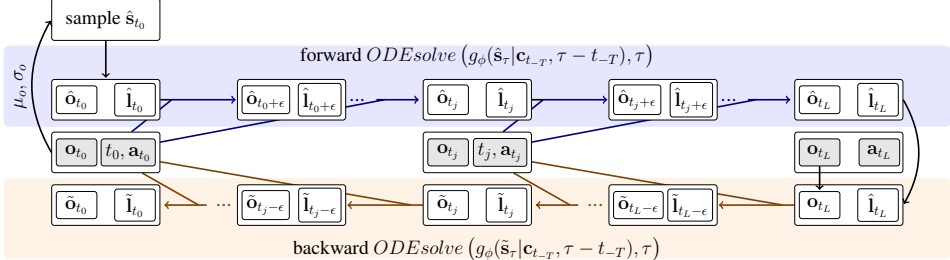

Figure 3: Full model structure for obtaining log-likelihood estimates. The flow of the log-likelihoods is not explicitly displayed, but in practice, it is concatenated to the twinned space $[\hat{l}_t, \hat{o}_t]$. We denote the forward run estimates $(t_0 \to t_L)$ with $\hat{\cdot}$ and the backward run estimates $(t_L \to t_0)$ with $\tilde{\cdot}$.

This is a key technical result that enables us to achieve precise log-likelihood estimates using the twinned space (detailed proof in the supplementary section D), and thus forms a pivotal component of the TIF framework. We later demonstrate the merits of log-likelihood estimation for outlier detection and as a confidence measure of the proposed model.

In order to estimate the log-likelihood using the discrete-time observations $o_{t_j}$ ($t_j \in T$), we must sample $l_t$ and $o_t$ using our model. We provide details of the algorithm to obtain these estimates in the supplementary material, and an illustration in Figure 3. To obtain the log-likelihood of full trajectory $\mathbf{H}_{t_L}$ the flow needs to be run backward ($t \to 0$) separately for each $o_{t_j}$ in $\mathbf{H}_{t_L}$. And the final estimate is obtained with Monte Carlo (MC) method. The stability of the MC estimate is an interesting question for future investigations. Computationally, this can be too expensive, so we instead use mean squared error (MSE) as our loss function during training. The time complexity of TIF is further discussed in the supplementary materials.

**Combining $\mathcal{D}_{obs}$ and $\mathcal{D}_{int}$:** Collecting interventional data $\mathcal{D}_{int}$ can be expensive or raise ethical concerns, motivating the use of additional observational data $\mathcal{D}_{obs}$. However, incorporating observational data is non-trivial in causal settings since the model can learn to replicate confounding effects that could lead to suboptimal performance. Previous studies (Ilse et al., 2021; Gasse et al., 2021; Manski, 1989) have established that careful inclusion of observational data can help in learning the dynamics for discrete interventional settings. We now present the first such results for more-challenging continuous-time settings with real-valued variables and partial observations.

**Corollary 3.2.** *TIF's estimator $\hat{p}_\phi(o_{t_{j+1}}|m_{t_{j+1}}, \mathbf{H}_{t_j}, do(a_{t_j}), i = 1)$ learned with $|\mathcal{D}_{obs}| \to \infty$ obtains strictly better generalization in $\mathcal{D}_{int}$ than if trained with $|\mathcal{D}_{obs}| = 0$.*

Full proof is given in the supplementary materials (Section E). This result motivates the use of observational data in model learning. It is worth noting that in practice the amount of observational data is limited, so we also experimentally demonstrate the benefits of combining $\mathcal{D}_{obs}$ with $\mathcal{D}_{int}$.

### 3.3 MITIGATING STIFFNESS

During the development of TIF, we faced the challenge of stiff ODEs, where the backward run trajectory deviated from the forward run (refer to Figure 4f). As a result, the initial Gaussian distribution couldn't be recovered, making it impossible to estimate the log-likelihood. Stiffness has been addressed in the literature through techniques like maintaining checkpoints on the forward run and using them in the reverse pass (Gholaminejad et al., 2019). However, these approaches are not applicable to our context, as we lack knowledge of the exact forward trajectories.

To counter this issue, we propose a novel regularization term $R(\hat{s}, \tilde{s})$ which penalizes the mean squared error (MSE) between the forward predicted states $\hat{s}_{t_j}$ and the corresponding backward predicted states $\tilde{s}_{t_j}$ at each time step $t_j \in T$

$$R(\hat{s}, \tilde{s}) = \frac{1}{L} \sum_{t_j \in T} (\hat{s}_{t_j} - \tilde{s}_{t_j})^2 \,, \tag{2}$$

The forward and backward trajectories coincide when no numerical error is incurred. Experimentally, we observed significant improvements with this regularization.

## 4 EXPERIMENTS

We performed detailed experiments to validate the efficacy of TIF. First, we showcase TIF's versatility in various tasks in a controlled setting by conducting simulations with a pendulum system. Specifically, we demonstrate TIF's capacity to combine observational and interventional data, its ability to handle masked observations, and its proficiency in log-likelihood estimation. We also demonstrate the benefits of the regularization penalty in handling stiffness, and put TIF to the test in the realm of counterfactual prediction. These experiments are designed to illustrate the flexibility of TIF, whereafter we move on to compare TIF's competitiveness in two standard benchmarks from reinforcement learning and medical domains. Specifically, we evaluate TIF's potential for treatment effect prediction using benchmark tumor growth data. In the other task, we compare TIF's performance with other established methods in the widely recognized Half Cheetah environment, a prominent benchmark within the RL field (Du et al., 2020; Feinberg et al., 2018; Buckman et al., 2018).

### 4.1 PENDULUM

We begin with a simple pendulum setup similar to De Brouwer et al. (2022), where continuous interventions influence the acceleration of the pendulum. The observable variables are the horizontal and vertical components of the pendulum angle: $\theta_x = sin(\theta)$ and $\theta_y = cos(\theta)$, while unobserved confounders are velocity $v$ and a pendulum length $l$ which is randomly sampled for each run. Interventions in $\mathcal{D}_{int}$ are defined by interventional policy $\pi_{int}(\theta_x(t), \theta_y(t))$, and the actions in $\mathcal{D}_{obs}$ are defined by a privileged policy $\pi_{prv}(\theta_x(t), \theta_y(t), l, v(t))$. A more detailed description of the simulation procedure and model training is given in the supplementary material.

**Stiffness** We encountered a clear stiffness problem in the system. Without the regularization term, the backward trajectory diverges from the forward run trajectory (Figure 4f). However, by introducing the penalty (Equation 2), stable invertible dynamics can be learned. During the first 100 training epochs, the RMSE between forward and backward trajectories increased to over 100 without regularization and remained under $10^{-3}$ with it (Figure 4e). We also investigated Jacobian and kinetic regularization (Finlay et al., 2020) but they turned out to be ineffective in our experiments.

**Combining $\mathcal{D}_{obs}$ and $\mathcal{D}_{int}$** We assess the model performance with three distinct training configurations: utilizing only $\mathcal{D}_{obs}$, only $\mathcal{D}_{int}$, or both. We conducted this experiment with 1000 observational trajectories and varying numbers of interventional samples. The outcomes using 10 different random seeds are illustrated in Figure 4d.

We note from Figure 4d that combining $\mathcal{D}_{int}$ and $\mathcal{D}_{obs}$ in training improves the performance, especially when the number of interventional samples is small. With 50 interventional samples, the combined training approach demonstrates a remarkable over 30% reduction in RMSE compared to the individual training methods. As the number of interventional samples is increased the performance of $\mathcal{D}_{int}$ coincides with the performance of combined training.

**Masking** TIF can seamlessly accommodate missing and partially masked observations. We demonstrate this in Figure 4c: even when 20% of the inputs are masked out, the model is still able to learn the dynamics reasonably. In general, we found that the root mean square error (RMSE) in the test set approximately triples when 20% of the observations are hidden.

**Outlier detection with log-likelihood estimate** As mentioned earlier, the log-likelihood estimate can still offer valuable insights into the data. For instance, it can help with detecting anomalies and unexpected side effects, e.g., in medical trials This is illustrated in Figure 4a, where part of the observations are corrupted. The negative log-likelihood of the validation data decreases during training as expected, and reflects how well the model fits the data (details in Supplementary).

**Counterfactual prediction** Given a factual trajectory $\mathbf{H}_{t_L}$ with an output $\boldsymbol{o}_{t_L}$, we wish to answer *what if* queries: what would the outcome have been if at time $t_j$ another action $\boldsymbol{a}'_{t_j}$ had taken place instead of the factual $\boldsymbol{a}_{t_j}$. We present the details of our counterfactual prediction algorithm in the supplementary materials (Section H).

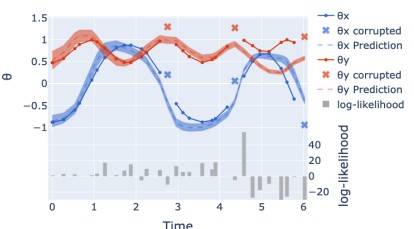 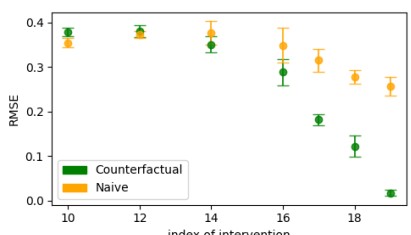

(a) The first corrupted observation can be detected based on the low log-likelihood shown at the bottom of the Figure, after that the estimates seem to recover. However, the log-likelihood estimates are not reliable after the next, due to conditioning on false information

(b) RMSE at the final state of the counterfactual trajectory using both counterfactual and naive predictions. The x-axis represents the intervention index, within the time points. The counterfactual prediction algorithm outperforms the naive approach.

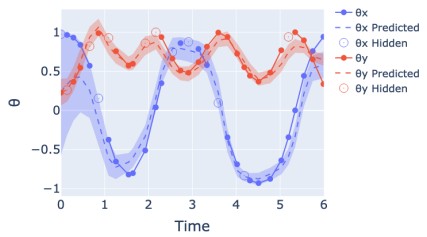 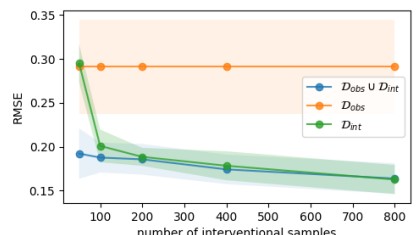

(c) The performance of TIF remains credible, even with 20% of the observations hidden. Even when part of the initial state is masked.

(d) Including observational data for training, i.e., using $(\mathcal{D}_{obs} \cup \mathcal{D}_{int})$ improves model performance especially when the number of interventional samples is small.

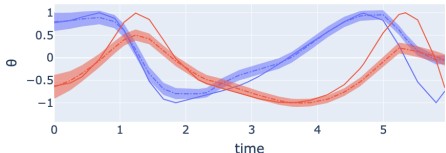 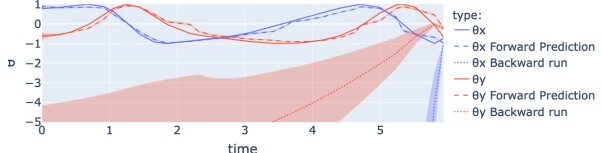

(e) With the proposed regularization, the backward and forward trajectories align.

(f) When trained without regularization the backward trajectories immediately diverge from the forward prediction.

Figure 4: Experiments on the pendulum environment.

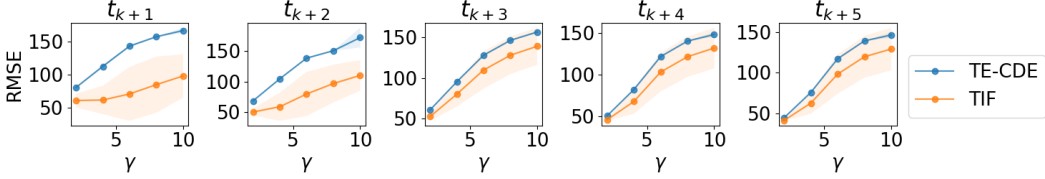

Figure 5: TIF exhibits superior performance in tumor growth prediction with strog confounding with $\kappa = 10$. The difference is particularly evident in short-term predictions $t_{k+1}$ and $t_{k+2}$.

To validate the efficacy of TIF for counterfactual prediction, we conducted experiments using a truncated pendulum dataset ($|T| = 20$), which included only a single intervention. This dataset comprised both factual (using $\pi_{obs}$) and counterfactual (using $\pi_{int}$) trajectories. For each pair of factual and counterfactual trajectories, we applied the counterfactual prediction algorithm as well as a naive online approach that predicted the counterfactual trajectory without incorporating information from the factual trajectory. We adjusted the index of the intervention and computed the RMSE at the final index (20). Figure 4b shows the results obtained with five random seeds. The counterfactual algorithm improves the prediction of the final state especially when the intervention takes place during later stages of the trajectory.

## 4.2 TUMOR DATA

For our second experiment, we benchmark on lung cancer tumor growth data under a Pharmacokinetic-Pharmacodynamic (PK-PD) setting (Geng et al., 2017). Specifically, a Hawkes process with self-exciting intensity function (Lee et al., 2016) is applied to obtain confounded observation times. Parameter $\gamma$ regulates the amount of confounding in data and $\kappa$ regulates the confounding in the Hawkes process (see Seedat et al. (2022) for details). We adopt the evaluation procedure for intervention prediction suggested for TE-CDEs Seedat et al. (2022) under distributional shift due to confounding.

We compare TIF with TE-CDEs in both strong ($\kappa = 10$) and moderate confounding scenarios ($\kappa = 5$). Observations that were masked in the Hawkes process were excluded from RMSE calculations. Figure 8 shows TIF's performance with strong confounding. Moderate confounding can be found from the supplementary materials.

## 4.3 HALF CHEETAH

For our final experiment, we assess the performance of TIF on the widely recognized Half Cheetah environment, a prominent benchmark for reinforcement learning settings (Du et al., 2020; Feinberg et al., 2018; Buckman et al., 2018). In the Half Cheetah task, the objective revolves around orchestrating the movement of joints for a cat-like robot (Wawrzyński, 2009) in order to achieve a forward running motion in a 2-dimensional space.

We combined model predictive control with the actor-critic methodology, utilizing deep deterministic policy gradient (Lillicrap et al., 2016) to learn the policy. Following (Du et al., 2020), we sample the environment with non-discrete time steps and train the model with an identical process to theirs. Within this experiment, TIF is employed to predict the future states, enabling the agent to make more informed decisions. We refer the reader to (Du et al., 2020) for a detailed description of the environment and training procedure.

We compare the performance of TIF against three strong baselines from literature, namely, Latent-ODE, RNN, and VAE-RNN. Figure 6 illustrates the evolution of mean test reward for different methods with increase in the number of environment steps. TIF demonstrated strong performance in terms of the mean test reward, which shows its promise for RL applications.

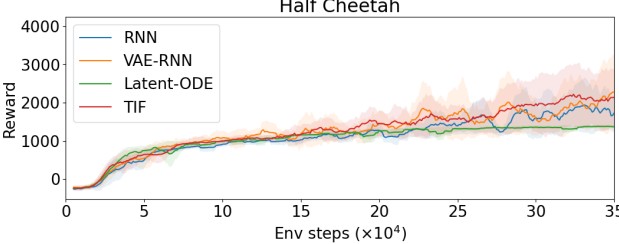

Figure 6: TIF achieves strong performance on the Half Cheetah RL benchmark. The shaded area represents the standard error calculated over 5 random seeds.

## 5 BROADER IMPACT AND LIMITATIONS

The idea of twinning proposed here to integrate observed and latent variables, with provision to include offline data, opens new avenues for dynamic system modeling and counterfactual inference. Thus it should be broadly useful in generative modeling, causal reasoning, and reinforcement learning settings.

TIF performs well on treatment effect prediction and reinforcement learning tasks, estimating log-densities. However, likelihood estimation can be computationally intensive, and its stability needs further study. Twinning resorts to modeling the system in a higher dimensional space, where confounders can be taken into account. This makes it suitable mainly for problems characterized by observation spaces with low or moderate dimension.

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

# Supplemental Materials: Twinned Interventional Flows

## A    NOTATION

Table 3: Overview of the notation used in the paper

| Notation | Explanation |
|---|---|
| $t$ | time |
| $t_j$ | discrete time point |
| $T$ | sequence of discrete time points |
| $t_{-T}$ | previous discrete time point in $T$ |
| $\boldsymbol{s}_t$ | state of the system at time $t$ |
| $\boldsymbol{o}_{t_j}$ | observation at discrete time point $t_j$ |
| $\boldsymbol{l}_{t_j}$ | latent variables at time $t_j$ |
| $\boldsymbol{a}_{t_j}$ | discrete action at time $t_j$ |
| $\boldsymbol{m}_{t_j}$ | mask of observation $o_{t_j}$ |
| $i$ | indicator of data source |
| $\mathbf{H}_{t_j}$ | history of observations at $t_j$ |
| $\boldsymbol{c}_{t_j}$ | context $[a_{t_j}, o_{t_j}, m_{t_j}, i]$ at time $t_j$ |
| $Q_{t_j,t}$ | transition between states $s_{t_j}$ and $s_t$ |
| $p(\cdot)$ | underlying probabilities |
| $q(\cdot)$ | time derivative of $s_t$ |
| $g_\phi(\cdot|\cdot)$ | conditioned CNF |
| $\phi$ | model parameters |
| $\hat{}$ | model approximations |
| $\tilde{}$ | backward run approximation |
| $f(\cdot)$ | maps discrete action to continuous time |
| $\mathcal{D}_{obs}$ | observational data |
| $\mathcal{D}_{int}$ | interventional data |
| $\pi_{prv}$ | privileged policy in $\mathcal{D}_{obs}$ |
| $\pi_{int}$ | policy in $\mathcal{D}_{int}$ |

Besides the *do* notation, we will also use an indicator $i$ to inform, whether the data is interventional or observational. Specifically, $i = 1$ pertains to interventions $do(\boldsymbol{a})$, whereas $i = 0$ is used to indicate that $\boldsymbol{a}$ is observed.

## B    MODEL ARCHITECTURE

We apply gate-bias modulation called *concatsquach* block (Abdal et al., 2021), which is illustrated in Figure 7.

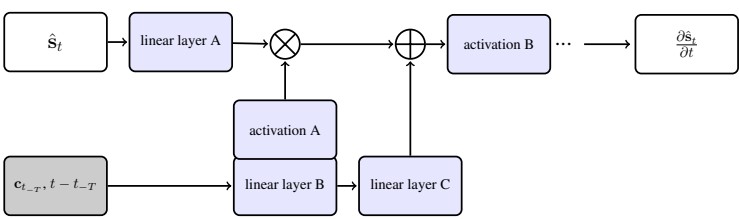

Figure 7: Concatsquach block, $\otimes$ denotes element wise multiplication and $\oplus$ element wise addition. Observable variables are marked with grey, and neural network layers with blue.

In our experiments on the pendulum environment and treatment effect prediction, 3 consecutive Concatsquach blocks were combined to obtain the time derivative. The dimension between the blocks is set to 128 and the output of the final block corresponds to $\hat{s}_t$ (see Figure 7). The sizes of linear layers B and C match the dimensions of $[\boldsymbol{c}_{t_{-T}}, t - t_{-T}]$ and the output size of the

corresponding linear layer A, respectively. The activation function A utilized the sigmoid function, and the activation function B employed hyperbolic tangent (tanh).

In the Half Cheetah environment, we used only 2 consecutive Concatsquach blocks with an in-between dimension of 128. The size of the model was reduced for fair comparison with the other methods.

The CNF is solved using python package *torchdiffeq* Chen (2021) with a fifth-order Dormand–Prince method ('*dopri5*') and the adjoint method (Chen et al., 2018).

As several feed-forward neural networks are Lipschitz continuous (Gouk et al., 2021), the architecture of $g_\phi$ can be modified with a reasonable degree of flexibility.

## C    Assumptions

**Markovian property**    By definition, POMDP satisfies the Markovian assumption with respect to $s_t$ (Bennett & Kallus, 2021). As we assume $o_{t_j}, l_{t_j}$ to be sufficient to describe $s_{t_j}$ fully, Markovian assumption also holds with respect to $o_{t_j}, l_{t_j}$. The Markovian causal model is illustrated in Figure 1 and it satisfies the parental Markov condition; that is, *each variable is $X_i$ is independent on all its nondescendants, given its parents* (Pearl & Verma, 1995).

Using the parental Markov condition (Pearl & Verma, 1995), we can describe the probability of a data sequence $\mathbf{H}_{t_L}$ as

$$p(\mathbf{H}_{t_L}|i) = p(\boldsymbol{m}_{t_0}, \boldsymbol{o}_{t_0}|i) \prod_{j=0}^{L-1} p(\boldsymbol{a}_{t_j}, \boldsymbol{m}_{t_{j+1}}, \boldsymbol{o}_{t_{j+1}}|\mathbf{H}_{t_j}, i) \tag{3}$$

$$= p(\mathbf{H}_{t_0}|i) \prod_{j=0}^{L-1} p(\boldsymbol{a}_{t_j}|\mathbf{H}_{t_j}, i)p(\boldsymbol{m}_{t_{j+1}}|\mathbf{H}_{t_j}, \boldsymbol{a}_{t_j}, i)p(\boldsymbol{o}_{t_{j+1}}|\boldsymbol{m}_{t_{j+1}}, \mathbf{H}_{t_j}, \boldsymbol{a}_{t_j}, i), \tag{4}$$

where the term $p(\boldsymbol{a}_{t_j}|\mathbf{H}_{t_j}, i)$ depends on the policy of the agent, $p(\boldsymbol{m}_{t_{j+1}}|\mathbf{H}_{t_j}, \boldsymbol{a}_{t_j}, i)$ is some masking process, and $p(\boldsymbol{o}_{t_{j+1}}|\boldsymbol{m}_{t_{j+1}}, \mathbf{H}_{t_j}, \boldsymbol{a}_{t_j}, i)$ arises from the underlying system dynamics.

Markovian assumption restricts ability of the model to capture long-term dependencies and complex temporal dynamics. We can overcome this limitation by increasing the dimensionality of the latent variables $l_{t_j}$ to incorporate additional information from the past. By expanding the latent space, we can potentially capture more historical context and extend the model's ability to account for longer-term dependencies. The trade-off lies in finding an appropriate balance between expressiveness and computational efficiency.

**Conditional independencies**    According to the Markovian causal model in Figure 1 and using the Markovian assumption, we can derive the following important conditional independencies

- $\boldsymbol{a}_{t_j}, \boldsymbol{m}_{t_j}, \boldsymbol{o}_{t_j}, \boldsymbol{s}_{t_{j+1}} \perp \boldsymbol{s}_{t<t_j}|\boldsymbol{s}_{t_j}, \mathbf{H}_{t_{j-1}}, \boldsymbol{a}_{t_{j-1}}$ (5)
- $\boldsymbol{s}_{t_j} \perp do(\boldsymbol{a}_{t_j})|\mathbf{H}_{t_j}, i = 1$ (6)
- $\boldsymbol{s}_{t_{j+1}}, \boldsymbol{o}_{t_{j+1}} \perp i|\boldsymbol{s}_{t_j}, \boldsymbol{a}_{t_j}, \mathbf{H}_{t_j}$ , (7)

where $\perp$ denotes the conditional independency given $|\cdot$ variables. The conditional independence is a concept in probability theory that captures the independence between random variables given the knowledge of other random variable, i.e., $A \perp B|C \Leftrightarrow P(A, B|C) = P(A|C)P(B|C)$.

Equation 5 follows from the parental Markov condition, as the current state together with historical observations and actions cover the parents of the future states. Similarly, Equation 6 indicates that the system state is conditionally independent of the current intervention, given the history and the fact that the data is drawn from the interventional scenario. This aligns with the parental Markov condition in the interventional regime. Lastly, Equation 7 expresses that the next state and observation are not dependent on whether the data is from interventional ($i = 1$) or observational ($i = 0$) scenario, as long as the current state, action, and observed history are known. This independence result holds irrespective of the data source, given that the underlying dynamic system is the same regardless of the action policy.

**Other assumptions** In addition to the Markovian property we need to make a few more assumptions for the theoretical results to hold.

**Assumption C.1.** *(Overlap) We assume that probabilities of all possible actions and observations in $\mathcal{D}_{int}$ also have non-zero probability in $\mathcal{D}_{obs}$. (Johansson et al., 2019; Seedat et al., 2022).*

**Assumption C.2.** *(Continuity) $g_\phi$ is uniformly Lipschitz continuous in $s_t$ and continuous in time intervals $[t_j, t_{j+1}]$ (Chen et al., 2018), i.e., dividing the flow into Lipschitz continuous sub-flows.*

**Assumption C.3.** *(Independent initial state) $\hat{s}_0$ follows a simple Gaussian prior $\hat{s}_0 \sim \mathcal{N}(\hat{s}_0|\mu, \sigma)$, such that each element of $\hat{s}_0$ is independently determined.*

**Identifiability** Achieving accurate predictions relies on the identifiability of true causal effects. Identifiability can be attained through additional assumptions, such as (weighted) time-independent sampling inspired by Tennenholtz et al. (2019), although this approach faces challenges related to the curse of dimensionality. An alternative, more tractable path involves a reduction to proximal causal inference, as outlined by (Bennett & Kallus, 2021). The specific assumptions for identifiability vary depending on the task and its features, and while we refrain from defining these here, we assume that the problems are identifiable, and observable variables suffice for accurate predictions of the full state.

## D LOG-LIKELIHOOD ESTIMATION

Chen et al. (2018) suggested a method for obtaining log-likelihood estimates and a maximum likelihood approach for the entire state space. However, we introduce an additional layer of complexity by incorporating unobserved but relevant latent variables into the twinned space. These latent variables are not directly observable. Consequently, we need to derive the correct formulation for estimating $\frac{\partial o_t}{\partial t}$. This derivation is essential as it allows us to estimate the log-likelihoods of the observed variables using a similar approach to Chen et al. (2018). Once we have established the correct form for $\log p(\boldsymbol{o}_{t_j}|\mathbf{H}_{t_j}, \boldsymbol{a}_{t_j}, i)$, we will demonstrate how this value can be estimated within the context of the TIF framework, leveraging its invertibility.

**Theorem D.1.** *The change in the log probabilities of observations in time can be expressed as an expected value of the trace over the latent variables $\hat{\boldsymbol{l}}_t$.*

$$\frac{\partial}{\partial t} \log(p(\boldsymbol{o}_t|\mathbf{H}_{t_{-T}}, \boldsymbol{a}_{t_{-T}}, i)) = \mathbb{E}_{p(\boldsymbol{l}_t|\boldsymbol{o}_t, \mathbf{H}_{t_{-T}}, \boldsymbol{a}_{t_{-T}}, i)} \left[ -Tr\left( \frac{\partial}{\partial \boldsymbol{s}} g_\phi(\boldsymbol{s}_t, |t - t_{-T}, \boldsymbol{c}_{t_{-T}}) \right) \right].$$

*Proof.* The proof builds on the general structure from Chen et al., while incorporating the latent variables of the twinned space. We use $\boldsymbol{o}$ to refer to the underlying continuous variable observed at the discrete observation times $T$. To clarify that we are here considering the underlying continuous variables we use the notation $o(t)$ instead of $\boldsymbol{o}_t$ and similarly $s(t)$ and $l(t)$. For convenience, we omit conditioning keeping it implicit since all probabilities are conditioned on $\mathbf{H}_{t_{-T}}, \boldsymbol{a}_{t_{-T}}, i$.

We denote transformation $(o(t), l(t)) \rightarrow (o(t + \epsilon), l(t + \epsilon))$ with $T_\epsilon(o(t), l(t))$. Now we can use the definition of derivative together with the change of variables formula to obtain

$$\frac{\partial}{\partial t} log p_o(o(t)) = \lim_{\epsilon \to 0^+} \frac{\log p_o(o(t + \epsilon)) - \log p_o(o(t))}{\epsilon}$$

$$= \lim_{\epsilon \to 0^+} \frac{\log \left( \int_{l(t+\epsilon) \in \mathcal{L}} p_{o,l}(o(t + \epsilon), l(t + \epsilon)) dl(t + \epsilon) \right) - \log p_o(o(t))}{\epsilon}$$

$$= \lim_{\epsilon \to 0^+} \frac{\log \left( \int_{l(t) \in \mathcal{L}} p_{o,l}(o(t), l(t)) |det \frac{\partial}{\partial s} T_\epsilon^{-1}(s(t))| dl(t) \right) - \log p_o(o(t))}{\epsilon},$$

where $det$ denotes the determinant of the transformation matrix.

We note that

$$\frac{\partial}{\partial t}logp_o(o(t)) = \lim_{\epsilon\to0^+} \frac{\log\left(\frac{1}{p_o(o(t))}\int_{l(t)\in\mathcal{L}}p_{o,l}(o(t),l(t))|det\frac{\partial}{\partial s}T_\epsilon^{-1}(s(t))|dl(t)\right)}{\epsilon}$$

$$= \lim_{\epsilon\to0^+} \frac{\log\left(\int_{l(t)\in\mathcal{L}}\frac{1}{p_o(o(t))}p_{o,l}(o(t),l(t))|det\frac{\partial}{\partial s}T_\epsilon^{-1}(s(t))|dl(t)\right)}{\epsilon}$$

$$= \lim_{\epsilon\to0^+} \frac{\log\left(\underbrace{\int_{l(t)\in\mathcal{L}}p_{l|o}(l(t)|o(t))|det\frac{\partial}{\partial s}T_\epsilon^{-1}(s(t))|dl(t)}_{\to1}\right)}{\epsilon} \ .$$

Now as the argument of the limit approaches the form $\frac{0}{0}$, we can apply L'Hôpital's rule by taking derivatives with respect to $\epsilon$ for both the numerator and denominator terms,

$$\frac{\partial}{\partial t}logp_o(o(t)) = \lim_{\epsilon\to0^+} \frac{\frac{\partial}{\partial\epsilon}\log\left(\int_{l(t)\in\mathcal{L}}p_{l|o}(l(t)|o(t))|det\frac{\partial}{\partial s}T_\epsilon^{-1}(s(t))|dl(t)\right)}{\frac{\partial}{\partial\epsilon}\epsilon}$$

$$= \lim_{\epsilon\to0^+}\frac{\partial}{\partial\epsilon}\log\left(\int_{l(t)\in\mathcal{L}}p_{l|o}(l(t)|o(t))|det\frac{\partial}{\partial s}T_\epsilon^{-1}(s(t))|dl(t)\right) \ .$$

Using the *log derivative trick*, that states that the derivative of log function can be written as $\frac{\partial}{\partial x}\log f(x) = \frac{\frac{\partial}{\partial x}f(x)}{f(x)}$, we get

$$\frac{\partial}{\partial t}logp_o(o(t)) = \lim_{\epsilon\to0^+}\frac{\frac{\partial}{\partial\epsilon}\int_{l(t)\in\mathcal{L}}p_{l|o}(l(t)|o(t))|det\frac{\partial}{\partial s}T_\epsilon^{-1}(s(t))|dl(t)}{\int_{l(t)\in\mathcal{L}}p_{l|o}(l(t)|o(t))|det\frac{\partial}{\partial s}T_\epsilon^{-1}(s(t))|dl(t)}$$

$$= \lim_{\epsilon\to0^+}\frac{\left(\int_{l(t)}p_{l|o}(l(t)|o(t))\frac{\partial}{\partial\epsilon}|det\frac{\partial}{\partial s}T_\epsilon^{-1}(s(t))|dl(t)\right)}{\underbrace{\left(\mathbb{E}_{p_{l|o}}|det\frac{\partial}{\partial s}T_\epsilon^{-1}(s(t))|\right)}_{\to1}} \ .$$

We assume that $s(t)$ is bounded, which implies that $g_\phi$, $T_\epsilon$, and $\frac{\partial T_\epsilon}{\partial s}$ are also bounded. Furthermore $p_{l|o}(l(t)|o(t))$ is assumed to be bounded. Thus the dominated convergence theorem allows us to interchange the order of limit and integral (Kamihigashi, 2020), and we have

$$\frac{\partial}{\partial t}logp_o(o(t)) = \int_{l(t)\in\mathcal{L}}p_{l|o}(l(t)|o(t))\lim_{\epsilon\to0^+}\frac{\partial}{\partial\epsilon}|det\frac{\partial}{\partial s}T_\epsilon^{-1}(s(t))|dl(t) \ .$$

Now we invoke Jacobi's formula

$$\frac{\partial}{\partial t}detA(t) = Tr\left(adj(A(t))\frac{\partial A(t)}{\partial t}\right) \ , \tag{8}$$

where $adj$ is the adjugate of the matrix. Equation 8 can be seen as a special case of the Fokker-Planck Equation for zero diffusion (Risken, 1996; Chen et al., 2018).

Using Equation 8, we write

$$\frac{\partial}{\partial t} log p_o(o(t)) = \int_{l(t)\in\mathcal{L}} p_{l|o}(l(t)|o(t))$$

$$\cdot Tr\left(\lim_{\epsilon\to 0^+}\left(\underbrace{adj(\frac{\partial}{\partial s}T_\epsilon^{-1}(s(t)))}_{\to 1}\frac{\partial}{\partial\epsilon}\frac{\partial}{\partial s}T_\epsilon^{-1}(s(t))\right)\right) dl(t)$$

$$= \int_{l(t)\in\mathcal{L}} p_{l|o}(l(t)|o(t))Tr\left(\lim_{\epsilon\to 0^+}\frac{\partial}{\partial\epsilon}\frac{\partial}{\partial s}T_\epsilon^{-1}(s(t))\right) dl(t) \ .$$

Finally, we can appeal to the Taylor series. It is important to note, that we assume $g_\phi$ to be expressive enough and $\hat{s}_t$ to have sufficiently large dimensions, such that $g_\phi(\hat{s}_t|c_{t_{-T}}, t - t_{-T})$ is an unbiased estimator of $T_\epsilon$. Using Taylor's approximation of $T_\epsilon$ we get

$$\frac{\partial}{\partial t} log p_o(o(t))$$

$$= \int_{l(t)\in\mathcal{L}} p_{l|o}(l(t)|o(t))Tr\left(\lim_{\epsilon\to 0^+}\frac{\partial}{\partial\epsilon}\frac{\partial}{\partial s}(s(t) - \epsilon g_\phi(s(t)) - \mathcal{O}(\epsilon^2) - \mathcal{O}(\epsilon^3)...)\right) dl(t)$$

$$= \int_{l(t)\in\mathcal{L}} p_{l|o}(l(t)|o(t))Tr\left(\lim_{\epsilon\to 0^+}\frac{\partial}{\partial\epsilon}(I - \frac{\partial}{\partial s}\epsilon g_\phi(s(t)) - \mathcal{O}(\epsilon^2) - \mathcal{O}(\epsilon^3)...)\right) dl(t)$$

$$= \int_{l(t)\in\mathcal{L}} p_{l|o}(l(t)|o(t))Tr\left(\lim_{\epsilon\to 0^+}(-\frac{\partial}{\partial s}g_\phi(s(t)) - \mathcal{O}(\epsilon) - \mathcal{O}(\epsilon^2)...)\right) dl(t)$$

$$= \int_{l(t)\in\mathcal{L}} p_{l|o}(l(t)|o(t))Tr\left(-\frac{\partial}{\partial s}g_\phi(s(t))\right) dl(t)$$

$$= \mathbb{E}_{p_{l|o}(l(t)|o(t))} - Tr\left(\frac{\partial}{\partial s}g_\phi(s(t))\right) \ .$$

Finally we recover the conditioning variables $\mathbf{H}_{t_{-T}}, \boldsymbol{a}_{t_{-T}}$ and $i$, and return to the subscript notation $(\boldsymbol{o}_t)$

$$\frac{\partial}{\partial t}\log(p(\boldsymbol{o}_t|\mathbf{H}_{t_{-T}}, \boldsymbol{a}_{t_{-T}}, i))$$

$$= \mathbb{E}_{p(\boldsymbol{l}_t|\boldsymbol{o}_t, \mathbf{H}_{t_{-T}}, \boldsymbol{a}_{t_{-T}}, i)}\left[-Tr\left(\frac{\partial}{\partial \boldsymbol{s}}g_\phi(\boldsymbol{s}_t, |t - t_{-T}, \boldsymbol{c}_{t_{-T}})\right)\right] \ .$$

$\square$

The marginal distribution of the initial state at time $t_0$ can be written as the expectation over latent variables,

$$p_o(\boldsymbol{o}_{t_0}) = \mathbb{E}_{p_l(\boldsymbol{l}_{t_0})}\left[p_{o|l}(\boldsymbol{o}_{t_0}|\boldsymbol{l}_{t_0})\right] \ . \tag{9}$$

Since the conditions of Fubini's theorem are satisfied (follows from Assumption C.2) we can interchange the order of expectation and integral to obtain a formula for the change in log-likelihood

$$\Delta \log p_o(\boldsymbol{o}_t|\mathbf{H}_{t_{-T}}, \boldsymbol{a}_{t_{-T}}, i) \tag{10}$$

$$= \mathbb{E}_{p(\boldsymbol{l}_t|\boldsymbol{o}_t, \mathbf{H}_{t_{-T}}, \boldsymbol{a}_{t_{-T}}, i)}\left[\int_{t_0}^t -Tr\left(\frac{\partial}{\partial \boldsymbol{s}}g_\phi(\boldsymbol{s}_\tau, \tau - \tau_{-T}, \boldsymbol{c}_{\tau_{-T}})\right) d\tau\right] \ .$$

Thus by running the system backward in time, we can recover $(l_{t_0}, o_{t_0})$ and evaluate the observation log-likelihoods.

**Corollary D.2.** *Using theorem D.1, Equation 9, and 10 we get a formula for the log-likelihood of observation $\boldsymbol{o}_t$*

$$\log p_o(\boldsymbol{o}_t|\mathbf{H}_{t-T}, \boldsymbol{a}_{t-T}, i) = \log \left( \mathbb{E}_{p_l(\boldsymbol{l}_{t_0})} \left[ p_{o|l}(\boldsymbol{o}_{t_0}|\boldsymbol{l}_{t_0}) \right] \right)$$
$$+ \mathbb{E}_{p(\boldsymbol{l}_t|\boldsymbol{o}_t, \mathbf{H}_{t-T}, \boldsymbol{a}_{t-T}, i)} \int_{t_0}^t -Tr \left( \frac{\partial}{\partial \boldsymbol{s}} g_\phi(\boldsymbol{s}_\tau|\tau - \tau_{-T}, \boldsymbol{c}_{\tau_{-T}}) \right) d\tau$$

**Estimating log-likelihood** To estimate the log-likelihood of $\boldsymbol{o}_{t_j}$, we first need to sample from $\hat{\boldsymbol{l}}_{t_j} \sim p(\boldsymbol{l}_{t_j}|\mathbf{H}_{t_j}, \boldsymbol{a}_{t_j}, i)$ by running the flow from the initial time $(0 \to t_j)$. Then we combine the obtained sample $\hat{\boldsymbol{l}}_{t_j}$ with observation $\boldsymbol{o}_{t_j}$. To obtain the best estimates of the intermediate values $\boldsymbol{l}_t$ and $\boldsymbol{o}_t$ we run the resulting twinned system $\hat{\boldsymbol{s}}_{t_j} = [\hat{\boldsymbol{l}}_{t_j}, \boldsymbol{o}_{t_j}]$ backward in time with CNF. During this backward run, we track the dynamics of $\tilde{\boldsymbol{o}}_t$ and the probability $p(\tilde{\boldsymbol{o}}_t)$ simultaneously. The result of Corollary D.2 is estimated as a part of this process with

$$\log p_o(\boldsymbol{o}_{t_j}|\mathbf{H}_{t_j}, \boldsymbol{a}_{t_j}, i) = \log \left( \mathbb{E}_{p(\tilde{\boldsymbol{l}}_{t_0}|\tilde{\boldsymbol{o}}_{t_0})} \left[ p_o(\tilde{\boldsymbol{o}}_{t_0}) \right] \right)$$
$$+ \mathbb{E}_{p(\tilde{\boldsymbol{l}}_t|\tilde{\boldsymbol{o}}_t, \mathbf{H}_{t-T}, \boldsymbol{a}_{t-T}, i)} \int_{t_0}^t -Tr \left( \frac{\partial}{\partial \boldsymbol{s}} g_\phi(\tilde{\boldsymbol{s}}_\tau|\tau - \tau_{-T}, \boldsymbol{c}_{\tau_{-T}}) \right) d\tau$$

where the initial state at time $t_0$ is recovered by running the system backward in time and by using the Assumption C.3 we have $p(\tilde{\boldsymbol{l}}_{t_0}|\tilde{\boldsymbol{o}}_{t_0}) = p(\tilde{\boldsymbol{l}}_{t_0})$.

*Hutchinson's trace estimator* (Hutchinson, 1990) can be used in multidimensional setting (Grathwohl et al., 2019) to obtain estimate of the trace

$$Tr\{J_{g_\phi}(\boldsymbol{s}_t|t - t_{-T}, \boldsymbol{c}_{t_{-T}})\} = \mathbb{E}_v \left[ \mathbf{v}^T J_{g_\phi}(\boldsymbol{s}_t|t - t_{-T}, \boldsymbol{c}_{t_{-T}}) \mathbf{v} \right] , \quad (11)$$

where $\mathbf{v}$ is a random vector with zero mean and unit covariance.

Finally, the expected values are estimated with the Monte Carlo method. The full algorithm can be found in Section G.

**With masking** If only a subset $\boldsymbol{o}_{t_j}^m$ of $\boldsymbol{o}_{t_j} = [\boldsymbol{o}_{t_j}^m, \boldsymbol{o}_{t_j}^c]$ is observed at time $t_j$ the log-likelihood of $\boldsymbol{o}_{t_j}^m$ can be recovered using the Corollary D.2 with replacement of $\hat{\boldsymbol{l}}$ with $\hat{\boldsymbol{l}}' := [\hat{\boldsymbol{l}}, \hat{\boldsymbol{o}}^c]$ and $\boldsymbol{o}_{t_j}$ with $\boldsymbol{o}_{t_j}^m$ i.e. considering missing observations as latent variables and handling them as such.

# E    ASYMPTOTIC BOUNDS FOR THE TRUE TRANSITION MODEL IN THE CONTINUOUS SETTING

By leveraging the observational data, we aim to reduce the reliance on a large amount of interventional samples. Previous studies by Ilse et al. (2021) and Gasse et al. (2021) have demonstrated that observational data can indeed help in learning the system dynamics for the interventional case. Both of them also provide theoretical asymptotic bounds that can be seen as a special case of the work by Manski (1989). However, these studies only consider discrete-time settings, which do not apply here. Therefore, our objective is to generalize their bounds to more complex continuous-time settings with real-valued variables and masking. The proof adapts the general structure of Gasse et al. (2021) and makes use of the conditional Independence's of POMDP presented in Section C.

Now the do-notation (Pearl, 2009) helps us to distinguish observational and interventional policies. In particular, $do(\boldsymbol{a}_t = \boldsymbol{a})$ denotes that the action $\boldsymbol{a}_t$ at time $t$ is set to $\boldsymbol{a}$ according the policy $\pi_{int}$. Besides the $do$ notation, we will also use an indicator $i$ to inform, whether the data is interventional or observational. Specifically, $i = 1$ pertains to interventions $do(\boldsymbol{a})$, whereas $i = 0$ is used to indicate that $a$ is observed and generated by policy $\pi_{prv}$.

**Proposition E.1.** *We can model $\frac{\partial}{\partial t} \log p(\boldsymbol{s}_t|t - t_{-T}, \boldsymbol{c}_{t_{-T}})$ with TIF. If the number and dimensionality of latent variables is sufficiently large to preserve information about the unobserved confounders and the Markovian property holds, then we obtain an unbiased estimator $\hat{p}_\phi(\boldsymbol{o}_{t_{j+1}}|\boldsymbol{m}_{t_{j+1}}, \mathbf{H}_{t_j}, do(\boldsymbol{a}_{t_j}), i = 1)$ of $p(\boldsymbol{o}_{t_{j+1}}|\boldsymbol{m}_{t_{j+1}}, \mathbf{H}_{t_j}, do(\boldsymbol{a}_{t_j}), i = 1)$.*

*Proof.* As the true $p(\boldsymbol{o}_{t_{j+1}}|\boldsymbol{m}_{t_{j+1}}, \mathbf{H}_{t_j}, do(\boldsymbol{a}_{t_j}), i=1)$ is restricted by the POMDP, and we have a sufficiently expressive model to capture the effect of the latent confounders, we are guaranteed an unbiased estimator. To obtain this estimate Corollary D.2 can be used. □

Now let us generalize the asymptotic bounds (Manski, 1989; Gasse et al., 2021; Ilse et al., 2021) to our setting.

**Theorem E.2.** *Assume that $|\mathcal{D}_{obs}| \to \infty$. The learned causal model $\hat{p}_\phi$ in $\mathcal{D}_{int}$ is bounded as,*

$$\prod_{j=0}^{L-1} \hat{p}_\phi(\boldsymbol{o}_{t_{j+1}}|\boldsymbol{m}_{t_{j+1}}, \mathbf{H}_{t_j}, do(\boldsymbol{a}_{t_j}), i=1)$$

$$\geq \prod_{j=0}^{L-1} p(\boldsymbol{a}_{t_j}|\mathbf{H}_{t_j}, i=0)p(\boldsymbol{o}_{t_{j+1}}|\boldsymbol{m}_{t_{j+1}}, \mathbf{H}_{t_j}, \boldsymbol{a}_{t_j}, i=0) \,,$$

*and*

$$\prod_{j=0}^{L-1} \hat{p}_\phi(\boldsymbol{o}_{t_{j+1}}|\boldsymbol{m}_{t_{j+1}}, \mathbf{H}_{t_j}, do(\boldsymbol{a}_{t_j}), i=1)$$

$$\leq \prod_{j=0}^{L-1} p(\boldsymbol{a}_{t_j}|\mathbf{H}_{t_j}, i=0)p(\boldsymbol{o}_{t_{j+1}}|\boldsymbol{m}_{t_{j+1}}, \mathbf{H}_{t_j}, \boldsymbol{a}_{t_j}, i=0)$$

$$+ (1 - \prod_{j=0}^{L-1} p(\boldsymbol{a}_{t_j}|\mathbf{H}_{t_j}, i=0)) \cdot \prod_{j=0}^{L-1} p(\boldsymbol{o}_{t_{j+1}}|\boldsymbol{m}_{t_{j+1}}, \mathbf{H}_{t_j}, \boldsymbol{a}_{t_j}, i=0) \,.$$

*Proof.* Noting the conditional independencies equation 6 and equation 7, we can write the transition probability as,

$$p(\boldsymbol{o}_{t_{j+1}}|\boldsymbol{m}_{t_{j+1}}, \mathbf{H}_{t_j}, do(\boldsymbol{a}_{t_j}), i=1)$$

$$= \int_{\boldsymbol{s}_{t_{j+1}} \in \mathcal{S}} p(\boldsymbol{s}_{t_{j+1}}, \boldsymbol{o}_{t_{j+1}}|\boldsymbol{m}_{t_{j+1}}, \mathbf{H}_{t_j}, do(\boldsymbol{a}_{t_j}), i=1)$$

$$= \int_{\boldsymbol{s}_{t_j} \in \mathcal{S}} p(\boldsymbol{s}_{t_j}|\mathbf{H}_{t_j}, i=1) \int_{\boldsymbol{s}_{t_{j+1}} \in \mathcal{S}} p(\boldsymbol{s}_{t_{j+1}}, \boldsymbol{o}_{t_{j+1}}|\boldsymbol{s}_{t_j}, \boldsymbol{m}_{t_{j+1}}, \mathbf{H}_{t_j}, \boldsymbol{a}_{t_j}, i=0) \,,$$

where $\int_{\boldsymbol{s}_{t_{j+1}} \in \mathcal{S}}$ is marginalization over the full state space $\mathcal{S}$.

Now by using the conditional independence $\boldsymbol{s}_{t_0} \perp i|\mathbf{H}_{t_0}$ (Assumption C.3) we obtain

$$\prod_{j=0}^{L-1} p(\boldsymbol{o}_{t_{j+1}}|\boldsymbol{m}_{t_{j+1}}, \mathbf{H}_{t_j}, do(\boldsymbol{a}_{t_j}), i=1)$$

$$= \sum_{\boldsymbol{s}_{0 \to L} \in \mathcal{S}} p(\boldsymbol{s}_{t_0}|\mathbf{H}_{t_0}, i=0) \prod_{j=0}^{L-1} p(\boldsymbol{s}_{t_{j+1}}, \boldsymbol{o}_{t_{j+1}}|\boldsymbol{s}_{t_j}, \boldsymbol{m}_{t_{j+1}}, \mathbf{H}_{t_j}, \boldsymbol{a}_{t_j}, i=0) \,,$$

where $\sum_{\boldsymbol{s}_{0 \to L} \in \mathcal{S}}$ denotes the marginalization $\int_{\boldsymbol{s}_{t_0} \in \mathcal{S}} \int_{\boldsymbol{s}_{t_1} \in \mathcal{S}} \cdots \int_{\boldsymbol{s}_{t_H} \in \mathcal{S}}$ over the state space at the discrete observation times $T$

Let us introduce a dummy variable $\boldsymbol{a}'$

$$\prod_{j=0}^{L-1} p(\boldsymbol{o}_{t_{j+1}}|\boldsymbol{m}_{t_{j+1}}, \mathbf{H}_{t_j}, do(\boldsymbol{a}_{t_j}), i=1) = \sum_{\boldsymbol{a}'_{0 \to L-1} \in \mathcal{A}} \sum_{\boldsymbol{s}_{0 \to L} \in \mathcal{S}} p(\boldsymbol{s}_{t_0}|\mathbf{H}_{t_0}, i=0)$$

$$\cdot \prod_{j=0}^{L-1} p(\boldsymbol{a}'_{t_j}|\boldsymbol{s}_{t_j}, \mathbf{H}_{t_j}, i=0)p(\boldsymbol{s}_{t_{j+1}}, \boldsymbol{o}_{t_{j+1}}|\boldsymbol{s}_{t_j}, \boldsymbol{m}_{t_{j+1}}, \mathbf{H}_{t_j}, \boldsymbol{a}_{t_j}, \boldsymbol{a}'_{t_j}, i=0) \,,$$

where again $\sum_{\boldsymbol{a}'_{0 \to L-1} \in \mathcal{A}}$ is marginalization over the action space $\mathcal{A}$ at times $T$.

Now considering the case $\boldsymbol{a}'_{0 \to L-1} = \boldsymbol{a}_{0 \to L-1}$ with the Assumption C.1 we get the lower bound

$$\prod_{j=0}^{L-1} p(\boldsymbol{o}_{t_{j+1}} | \boldsymbol{m}_{t_{j+1}}, \mathbf{H}_{t_j}, do(\boldsymbol{a}_{t_j}), i = 1) \geq \prod_{j=0}^{L-1} p(\boldsymbol{a}_{t_j} | \mathbf{H}_{t_j}, i = 0) p(\boldsymbol{o}_{t_{j+1}} | \boldsymbol{m}_{t_{j+1}}, \mathbf{H}_{t_j}, \boldsymbol{a}_{t_j}, i = 0) .$$

Let $\wedge$ denote the logical AND operator. To provide the upper bound we isolate events $\boldsymbol{a}'_{t_0} \neq \boldsymbol{a}_{t_0}$, $(\boldsymbol{a}'_{t_0} = \boldsymbol{a}_{t_0}) \wedge (\boldsymbol{a}'_{t_1} \neq \boldsymbol{a}_{t_1})$, $(\boldsymbol{a}'_{t_0} = \boldsymbol{a}_{t_0}) \wedge (\boldsymbol{a}'_{t_1} = \boldsymbol{a}_{t_1}) \wedge (\boldsymbol{a}'_{t_2} \neq \boldsymbol{a}_{t_2})$, and so on, and write

$$\prod_{j=0}^{L-1} p(\boldsymbol{o}_{t_{j+1}} | \boldsymbol{m}_{t_{j+1}}, \mathbf{H}_{t_j}, do(\boldsymbol{a}_{t_j}), i = 1)$$

$$= \underbrace{\prod_{j=0}^{L-1} p(\boldsymbol{a}_{t_j} | \mathbf{H}_{t_j}, i = 0) p(\boldsymbol{o}_{t_{j+1}} | \boldsymbol{m}_{t_{j+1}}, \mathbf{H}_{t_j}, \boldsymbol{a}_{t_j}, i = 0)}_{\boldsymbol{a}'_{t_j} = \boldsymbol{a}_{t_j}}$$

$$+ \sum_{\boldsymbol{a}'_{0 \to L-1} \in A' \neq A} \sum_{\boldsymbol{s}_{0 \to L} \in S} p(\boldsymbol{s}_{t_0} | \mathbf{H}_{t_0}, i = 0)$$

$$\cdot \prod_{j=0}^{L-1} p(\boldsymbol{a}'_{t_j} | \mathbf{H}_{t_j}, \boldsymbol{s}_{t_j}, i = 0) p(\boldsymbol{o}_{t_{j+1}}, \boldsymbol{s}_{t_{j+1}} | \boldsymbol{m}_{t_{j+1}}, \boldsymbol{s}_{t_j}, \mathbf{H}_{t_j}, \boldsymbol{a}_{t_j}, \boldsymbol{a}'_{t_j}, i = 0)$$

$$= \prod_{j=0}^{L-1} p(\boldsymbol{a}_{t_j} | \mathbf{H}_{t_j}, i = 0) p(\boldsymbol{o}_{t_{j+1}} | \boldsymbol{m}_{t_{j+1}}, \mathbf{H}_{t_j}, \boldsymbol{a}_{t_j}, i = 0)$$

$$+ (1 - \prod_{j=0}^{L-1} p(\boldsymbol{a}_{t_j} | \mathbf{H}_{t_j}, i = 0)) \cdot \prod_{j=0}^{L-1} p(\boldsymbol{o}_{t_{j+1}} | \boldsymbol{m}_{t_{j+1}}, \mathbf{H}_{t_j}, \boldsymbol{a}_{t_j}, i = 0) .$$

By proposition E.1, with TIF we can obtain $\hat{p}_\phi(\boldsymbol{o}_{t_{j+1}} | \boldsymbol{m}_{t_{j+1}}, \mathbf{H}_{t_j}, do(\boldsymbol{a}_{t_j}), i = 1)$ that is an unbiased estimator of $p(\boldsymbol{o}_{t_{j+1}} | \boldsymbol{m}_{t_{j+1}}, \mathbf{H}_{t_j}, do(\boldsymbol{a}_{t_j}), i = 1)$. So the bound for the asymptotic setting given in the theorem statement follows immediately.

$\square$

**Corollary E.3.** *TIF's estimator $\hat{p}_\phi(\boldsymbol{o}_{t_{j+1}} | \boldsymbol{m}_{t_{j+1}}, \mathbf{H}_{t_j}, do(\boldsymbol{a}_{t_j}), i = 1)$ learned with $|\mathcal{D}_{obs}| \to \infty$ obtains strictly better generalization in $\mathcal{D}_{int}$ than if trained with $|\mathcal{D}_{obs}| = 0$.*

*Proof.* The corollary holds following the same argument as Gasse et al. (2021), which we reproduce here for completeness. There exists at least one $(\boldsymbol{a}_{t_j}, \mathbf{H}_{t_j})$ that has non zero probability in observation data, and furthermore there is some $\boldsymbol{o}_{t_{j+1}}$ for which

$$\prod_{j=0}^{L-1} p(\boldsymbol{a}_{t_j} | \mathbf{H}_{t_j}, i = 0) p(\boldsymbol{o}_{t_{j+1}} | \boldsymbol{m}_{t_{j+1}}, \mathbf{H}_{t_j}, \boldsymbol{a}_{t_j}, i = 0) > 0 .$$

Thus, by the lower bound $\prod_{j=0}^{L-1} \hat{p}_\phi(\boldsymbol{o}_{t_{j+1}} | \boldsymbol{m}_{t_{j+1}}, \mathbf{H}_{t_j}, do(\boldsymbol{a}_{t_j}), i = 1)$ is strictly positive. Furthermore the model $\hat{p}_\phi$ learned with both the interventional and the observational data introduces additional conditioning due to the observational data, and thus belongs to a class of estimators that is strictly subsumed by the one that contains the true transition model pertaining to the interventional data alone, so offers better generalization.

$\square$

Our contribution lies in extending the result to a fully continuous system with possible missing variables and masking.

## F  SIGNIFICANCE OF THE THEORETICAL RESULTS

We now summarize the motivation behind our theoretical results in Table 4.

| | Result | Importance and novelty |
|---|---|---|
| 1. | The change in the log probabilities of observations in time can be expressed as an expected value of the trace over the latent variables $\hat{l}_t$. (Theorem 3.2.) | This theorem enables precise log-likelihood estimation within the twinned space (Corollary 3.3). This forms a pivotal component of the TIF framework, facilitating the incorporation of Neural ODE benefits. A key technical challenge lies in the accurate formulation of the expected value over latent variables, which can be sampled with the model accordingly. |
| 2. | The performance of true transition model is bounded by the transition model learned from observational data | From this result we have that TIF's estimator $\hat{p}_\phi(\boldsymbol{o}_{t_{j+1}}|\boldsymbol{m}_{t_{j+1}}, \mathbf{H}_{t_j}, do(\boldsymbol{a}_{t_j}), i = 1)$ learned with $|\mathcal{D}_{obs}| \to \infty$ obtains strictly better generalization in $\mathcal{D}_{int}$ than if trained with $|\mathcal{D}_{obs}| = 0$. This motivates the use of observational data in model learning. These are the first such results in a fully continuous setting (exiting results apply only to discrete settings). Moreover, the result holds even with possible missing variables and masking, which to our knowledge, has not been dealt with previously. |

Table 4: Summary and significance of theoretical findings

## G  ALGORITHM

As mentioned before, using the log-likelihood as a loss function for training the model can be computationally intensive. Therefore, we utilize MSE for model training in the TIF framework. Consequently, TIFs can be run in two settings: the training setting (also used for simple forward prediction), outlined in Algorithm 1 and in log-likelihood estimation setting, see Algorithm 2.

The CNF is solved using an explicit numerical method, namely the fifth-order Dormand–Prince. To ensure accuracy, we set absolute and relative tolerances to $1e^{-4}$ during the integration process. To handle possible discontinuities of the derivative, we utilize the *torchdiffeq* package (Chen, 2021) with the keyword *jump_t*. By doing so, we eliminate the need for a for-loop structure (e.g., lines 8 and 16 in Algorithm 1) to achieve piece-wise integrations. Instead, we can call the ODE solver just once, making the computation more efficient.

Throughout our algorithms (see Algorithms 1 and 2), we refer to the *torchdiffeq* solver as *ODEsolve*. This solver takes the current state as input and employs the provided function to evaluate the derivative. Subsequently, the numerical solver handles the ODE integration between the specified time points. It's important to mention that the function evaluating the derivative has access to both the current context $\boldsymbol{c}_{t_T}$ and the time elapsed from the previous intervention $t - t_T$.

We ensure that the Assumption C.3 is followed by sampling the initial state according to line 6 in Algorithm 1. More precisely, the mean of each element of $\hat{\boldsymbol{o}}_0$ is set to the corresponding observation $\boldsymbol{o}_0$ while the means of latent variables are set to zero. The standard deviation of observed variables is set to a slightly smaller value than for the unobserved variables.

Assuming that we are interested in estimating the log-likelihood of $\boldsymbol{o}_{t_k}$ and have already obtained N samples of $\hat{\boldsymbol{s}}_{t_k}$, we can invoke the procedure described in Algorithm 2 and illustrated in Figure 3.

It is crucial to note that when the focus is on obtaining a log-likelihood estimate, the regularization term introduced in Equation 2 should be utilized. This term has no impact on the model if the assumption of an easily invertible system holds. Moreover, it serves to prevent nonsensical estimates that may arise from an unrecovered initial distribution. On the other hand, when only forward

---

**Algorithm 1** Model training and forward prediction

---

**input** observations $\boldsymbol{o}_{t_0}, \boldsymbol{o}_{t_1} \dots \boldsymbol{o}_{t_L}$, actions $\boldsymbol{a}_{t_0}, \boldsymbol{a}_{t_1} \dots \boldsymbol{a}_{t_L}$, masks $\boldsymbol{m}_{t_0}, \boldsymbol{m}_{t_1} \dots \boldsymbol{m}_{t_L}$, indicator $i$, times $T$

1: **function** dynamics$(\hat{\boldsymbol{s}}_t, \boldsymbol{c}_{t_{-T}}, i, t)$
2:     $\frac{\partial \hat{\boldsymbol{s}}_t}{\partial t} = g_\phi(\hat{\boldsymbol{s}}_{t_j} | \boldsymbol{c}_{t_{-T}}, t - t_{-T})$
3:     return $\frac{\partial \hat{\boldsymbol{s}}_t}{\partial t}$
4: **end function**
5: **repeat**
6:     Initialize $\hat{\boldsymbol{s}}_0 = [\hat{\boldsymbol{l}}_0, \hat{\boldsymbol{o}}_0]$, such that $\hat{\boldsymbol{o}}_0 \sim \mathcal{N}(\hat{\boldsymbol{o}}_0 | \mu(\boldsymbol{o}_0), \sigma(\boldsymbol{o}_0))$ and $\hat{\boldsymbol{l}}_0 \sim \mathcal{N}(\hat{\boldsymbol{l}}_0 | \boldsymbol{0}, I)$.
7:     Initialize $loss = 0, penalty = 0$
8:     **for** $t_j$ in $[t_0, t_1, t_2 \dots t_{L-1}]$ **do**
9:        set $\boldsymbol{c}_{t_{-T}} = [\boldsymbol{a}_{t_j}, \boldsymbol{o}_{t_j}, \boldsymbol{m}_{t_j}, i]$
10:       set $t_{-T} = t_j$
11:       $\hat{\boldsymbol{s}}_{t_{j+1}} = \text{ODEsolve}(\hat{\boldsymbol{s}}_{t_j}, dynamics, t_j, t_{j+1})$
12:       $loss += MSE(\boldsymbol{o}_{t_{j+1}}, \hat{\boldsymbol{o}}_{t_{j+1}})$
13:     **end for**
14:     **if** use regularization **then**
15:       Initialize $\tilde{\boldsymbol{s}}_{t_L} = \hat{\boldsymbol{s}}_{t_L}$.
16:       **for** $t_j$ in $[t_L, t_{L-1}, t_{L-2} \dots t_1]$ **do**
17:         set $\boldsymbol{c}_{t_{-T}} = [\boldsymbol{a}_{t_{j-1}}, \boldsymbol{o}_{t_{j-1}}, \boldsymbol{m}_{t_{j-1}}, i]$
18:         set $t_{-T} = t_{j-1}$
19:         $\tilde{\boldsymbol{s}}_{t_{j-1}} = \text{ODEsolve}(\tilde{\boldsymbol{s}}_{t_j}, dynamics, t_j, t_{j-1})$
20:         $penalty += MSE(\hat{\boldsymbol{s}}_{t_{j-1}}, \tilde{\boldsymbol{s}}_{t_{j-1}})$
21:       **end for**
22:     **end if**
23:     Calculate Loss: $(loss + penalty)/(L-1)$
24:     Update $g_\phi$ with gradient descent
25: **until** Training is done

---

**Algorithm 2** Log-likelihood estimation of $\boldsymbol{o}_{t_k}$

---

**input** observations $\boldsymbol{o}_{t_0}, \boldsymbol{o}_{t_1} \dots \boldsymbol{o}_{t_L}$, actions $\boldsymbol{a}_{t_0}, \boldsymbol{a}_{t_1} \dots \boldsymbol{a}_{t_L}$, masks $\boldsymbol{m}_{t_0}, \boldsymbol{m}_{t_1} \dots \boldsymbol{m}_{t_L}$, indicator $i$, times $T$, $N$ forward run state estimate samples $\hat{S}_{t_k} = [\hat{\boldsymbol{s}}_{t_k}^1, \hat{\boldsymbol{s}}_{t_k}^2 \dots \hat{\boldsymbol{s}}_{t_k}^N]$

1: **function** aug_dynamics$(\hat{\boldsymbol{s}}_t, \log p(\hat{\boldsymbol{s}}_t), \boldsymbol{c}_{t_{-T}}, i, t, \phi)$
2:     Initialize $\boldsymbol{v}$ from a Rademacher Distribution.
3:     $\frac{\partial \hat{\boldsymbol{s}}_t}{\partial t} = g_\phi(\hat{\boldsymbol{s}}_{t_j} | \boldsymbol{c}_{t_{-T}}, t - t_{-T})$
4:     $\frac{\partial log p(\hat{\boldsymbol{s}}_t)}{\partial t} = -\boldsymbol{v}^T J_{g_\phi}(\hat{\boldsymbol{s}}_{t_j} | \boldsymbol{c}_{t_{-T}}, t - t_{-T}) \boldsymbol{v}$
5:     return $[\frac{\partial \hat{\boldsymbol{s}}_t}{\partial t}, \frac{\partial log p(\hat{\boldsymbol{s}}_t)}{\partial t}]$
6: **end function**
7: **for** $\hat{\boldsymbol{s}}_{t_k}$ in $\hat{S}_{t_k}$ **do**
8:     set $\tilde{\boldsymbol{s}}_{t_k} = [\hat{\boldsymbol{l}}_{t_k}, \boldsymbol{o}_{t_k}]$
9:     set $\Delta log p(\tilde{\boldsymbol{s}}) = 0$
10:     **for** $t_j$ in $[t_k, t_{k-1}, t_{k-2} \dots t_1]$ **do**
11:       set $\boldsymbol{c}_{t_{-T}} = [\boldsymbol{a}_{t_{j-1}}, \boldsymbol{o}_{t_{j-1}}, \boldsymbol{m}_{t_{j-1}}, i]$
12:       set $t_{-T} = t_{j-1}$
13:       $[\tilde{\boldsymbol{s}}_{t_{j-1}}, \Delta \log p(\tilde{\boldsymbol{s}})] = \text{ODEsolve}(\tilde{\boldsymbol{s}}_{t_j}, \text{aug\_dynamics}, t_j, t_{j-1})$
14:     **end for**
15:     evaluate $\log p(\tilde{\boldsymbol{o}}_0)$ from the known initial distribution $\mathcal{N}(\mu(\boldsymbol{o}_0), \sigma(\boldsymbol{o}_0))$
16:     Save samples $\Delta \log p(\tilde{\boldsymbol{s}})$ and $\log p(\tilde{\boldsymbol{o}}_0)$
17: **end for**
18: Compute MC estimate of $log p(\boldsymbol{o}_{t_k} | \mathbf{H}_{t_k})$

---

prediction is desirable the regularization can be relaxed by replacing line 20 in Algorithm 1 with $MSE(\hat{\boldsymbol{o}}_{t_{j-1}}, \tilde{\boldsymbol{o}}_{t_{j-1}})$.

Another remark worth noting is that when TIF is trained according to Algorithm 1, it reduces to regular NODE. By discarding the flow of log-likelihoods in training and forward prediction we can gain faster performance.

## H    COUNTERFACTUAL PREDICTION

While the interventional query aims to marginalize over the latent variables, the counterfactual query conditions on them (Khemakhem et al., 2021). In other words, the goal is to model the distribution of $\boldsymbol{o}_{t_L}$ after action $do(\boldsymbol{a}_{t_j})$ given that $\boldsymbol{a}'_{t_j}$ and $\boldsymbol{o}'_{t_L}$ have in fact occurred. The counterfactual inference can be divided into three steps described in Theorem 7.1.7 by Pearl (2009, p. 206). Here we list these steps according to our context and notation,

1. **Abduction:** Use the evidence $\boldsymbol{o}'_{t_L}$ and $\boldsymbol{a}'_{t_j}$ to infer the conditional distribution over the latent variables $p(\boldsymbol{l}_{t_j}|\boldsymbol{o}'_{t_L}, \boldsymbol{a}'_{t_j})$.

2. **Action:** Introduce the action $do(\boldsymbol{a}_{t_j})$ based on the counterfactual query together with the obtained $p(\boldsymbol{l}_{t_j}|\boldsymbol{o}'_{t_L}, \boldsymbol{a}'_{t_j})$.

3. **Prediction:** Use the conditioned state to predict the distribution of $\boldsymbol{o}_{t_L}$ given action $do(\boldsymbol{a}_{t_j})$.

The first step, *abduction*, requires evaluating the posterior distribution of the latent variables $\boldsymbol{l}_{t_j}$ after observing the factual trajectory $(\boldsymbol{o}'_{t_L}, \boldsymbol{a}'_{t_j})$. This is illustrated with a blue solid line in Figure 2b. Flow-based models can naturally run both forward and backward in time, allowing us to obtain the posterior estimate $\tilde{\boldsymbol{l}}_{t_j} \sim p(\boldsymbol{l}_{t_j}|\boldsymbol{o}'_{t_L}, \boldsymbol{a}'_{t_j})$. This estimate is obtained by first forward running $t_0 \to t_L$, then replacing $\hat{\boldsymbol{o}}_{t_L}$ with $\boldsymbol{o}_{t_L}$, and finally running backward $t_L \to t_j$ to obtain a more informative $\tilde{\boldsymbol{l}}_{t_j}$. The backward run trajectory is shown in orange in Figure 2b.

The second step, *action*, can be introduced as an input to the model. Additionally, the observation $\boldsymbol{o}_{t_j}$ can be twinned with the obtained $\tilde{\boldsymbol{l}}_{t_j}$ from the first step to obtain better estimate ($\tilde{\boldsymbol{s}}_{t_j} = [\tilde{\boldsymbol{l}}_{t_j}, \boldsymbol{o}_{t_j}]$) of $\boldsymbol{s}_{t_j}$.

Finally, the *prediction* can be obtained by running the model as follows

$$\tilde{\boldsymbol{s}}_{t_L} = \tilde{\boldsymbol{s}}_{t_j} + \int_{t_j}^{t_L} g_\phi(\tilde{\boldsymbol{s}}_t|\cdot)dt . \tag{12}$$

This is shown with a green continuous line in Figure 2b. It is worth noting that the equation above assumes that there are no additional actions nor observations between $t_j$ and $t_L$. However, if there were some intermediate steps they could be included by running the model in parts.

## I    PENDULUM EXPERIMENTS

## J    TUMOR GROWTH EXPERIMENTS

### J.1    SIMULATION OF DATA

The system of differential equations determining the pendulum dynamics is

$$\frac{d\theta(t)}{dt} = v \tag{13}$$

$$\frac{dv(t)}{dt} = \left(1 + f(a_{t_{-T}}, t - t_{-T})\right)\left(\frac{-g}{l}\right)\sin(\theta(t)) \tag{14}$$

$$\frac{df(a_{t_{-T}}, t - t_{-T})}{dt} = a_{t_{-T}} - \delta e^{-\delta(t-t_{-T})} , \tag{15}$$

where $\delta$ is set to 1, and $g = 9.81$ corresponds to the acceleration due to gravity.

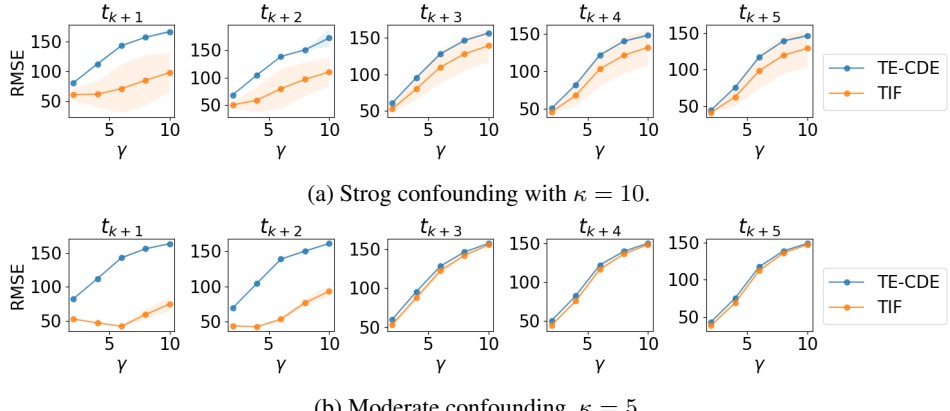

(a) Strog confounding with $\kappa = 10$.

(b) Moderate confounding, $\kappa = 5$.

Figure 8: TIF exhibits superior performance in tumor growth prediction across varying degrees of confounding. The difference is particularly evident in short-term predictions $t_{k+1}$ and $t_{k+2}$.

The length of the pendulum is randomly sampled as $l \sim \mathcal{U}(1.5, 3.5)$, where $\mathcal{U}(a, b)$ is a uniform distribution between $a$ and $b$. The initial angle of the pendulum is also randomly sampled as $\theta(0) \sim \mathcal{U}(0.5, 5.5)$, and the initial velocity $v(0)$ is set to zero. The observable variables are the $x$ and $y$ components of the pendulum angle: $\theta_x = sin(\theta)$ and $\theta_y = cos(\theta)$.

The data trajectories are simulated over 6 time units, and $H = 30$ irregular sample times in $T$ are drawn. Two random intervention times are sampled for each trajectory, one in the first half and one in the latter half, the effective action based on equation 15 is given as the action channel $a_{t_j}$ at each observation time in $T$ together with the $\theta_x(t_j), \theta_y(t_j)$.

The agent on observational data has privileged information on the length and velocity of the pendulum. We define an arbitrary privileged policy as,

$$\pi_{prv}(\theta_x(t), \theta_y(t), l, v(t)) \sim \mathcal{U}(0, 15) \cdot \frac{(l/4) \cdot \theta_y(t)}{|v(t)| + 1} , \qquad (16)$$

where the policy is responsible for deciding the action amplitude at those 2 intervention times.

For interventional policy, we define

$$\pi_{int}(\theta_x(t), \theta_y(t)) \sim \mathcal{U}(0, 15) \cdot \frac{(\theta - 0.9)_x(t)}{2}. \qquad (17)$$

To illustrate the distributional shift between $\pi_{prv}$ and $\pi_{int}$ we draw 200 samples from both policies and visualized the action amplitudes of both actions with respect to the pendulum angle. The results are shown in Figure 9.

### J.2 TRAINING WITH PENDULUM DATA

TIF learns the system dynamics in a 20-dimensional state space, consisting of 18 latent variables and 2 observational variables $(\theta_x, \theta_y)$. In all but the log-likelihood and counterfactual prediction experiments, the regularization was relaxed, by computing the MSE only on the observational variables $R(\hat{o}, \tilde{o})$, as the invertibility of the system is not needed in the other experiments.

The initial state was sampled with a Gaussian priors $\hat{o}_0 \sim \mathcal{N}(\hat{o}_0|o_0, 0.2)$ and $\hat{l}_0 \sim \mathcal{N}(\hat{l}_0|0, 1)$.

The model was trained for 300 (excluding the counterfactual experiment), using an adaptive learning rate Adam (Kingma & Ba, 2015) with a maximum learning rate of $0.005$. The model used for testing was selected based on the performance on the validation set. To illustrate the predictions made by TIF and the effect of initial state we draw 20 trajectories (Figure 10).

With the full regularization, the same level of RMSE in test data as with relaxed regularization was obtained within 400 epochs, while using relative and absolute tolerances of $10^{-8}$. Thus

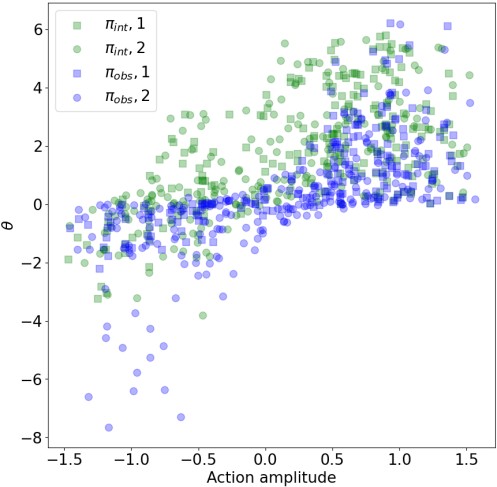

Figure 9: Distributional shift between observational and interventional data from two fixed intervention times.

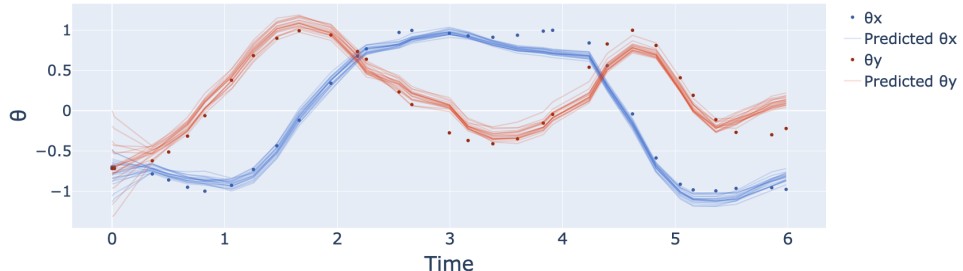

Figure 10: To illustrate the generative nature of TIF and to visualize its performance in pendulum data, 20 initial states were sampled and corresponding trajectories are shown together with observations.

regularization slows the training process but enables learning an invertible model that is equally expressive. The negative log-likelihood estimate also decreases during training in validation data, as expected.

In Figure 11 we illustrate how the log-likelihood estimate acts at the beginning of training (after one epoch). As expected the log-likelihood reflects the distance of observations from the trivial predictions.

**Counterfactual prediction** For the counterfactual prediction task, the model was trained using 800 samples from $\mathcal{D}_{obs}$ and 400 samples from $\mathcal{D}_{int}$, for a maximum of 600 epochs. Also, the full regularization was applied.

# K  HYPER-PARAMETER TUNING

For the pendulum data, hyper-parameters such as the latent dimension were fine-tuned, and the same configuration was applied to the tumor growth experiment. We acknowledge that more extensive hyperparameter tuning could potentially lead to further improvements. The learning rate was determined through multiple runs with a small number of epochs.

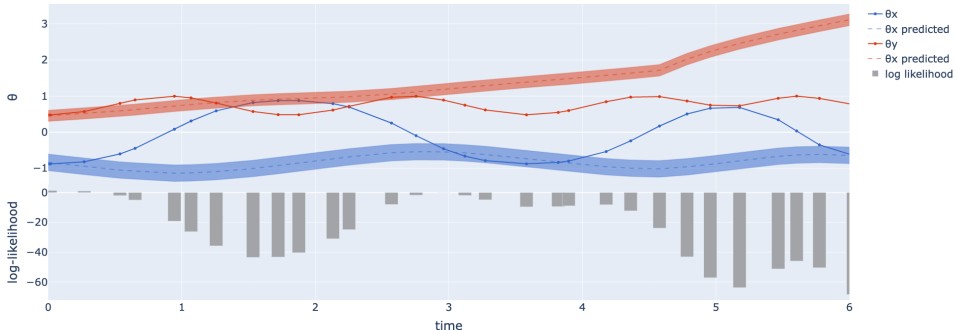

Figure 11: At the beginning of training log-likelihoods clearly reflect the distance of observations from the trivial model predictions

In the case of the RL benchmark, the latent dimensionality and model size were aligned with those of other benchmarks, with the latent dimension being an educated guess based on the dimensions of the Half Cheetah task. The primary parameter tuned in this scenario was the learning rate, with approximately five different options tested for each setup.

## L    TIME COMPLEXITY

TIF inherits from the continuous flows normalizing a more efficient way of estimating the log likelihood.

If the cost of evaluating TIF is $O(DH)$ where $D$ is the dimensionality of the data, and $H$ is the largest hidden dimension in the concatsquach block, then the time complexity of one transformation using a naive normalizing flow approach would be $O(DH + D^3)$

The cubic format arises from using the change of variables to obtain complex densities $z \sim p_z(z)$ from a simple distribution $u \sim p_u(u)$ with transformation $T(u) = x$, as described in (Papamakarios et al., 2021),

$$log(p_z(z)) = log(p_u(u)) - log(|det J_T(u)|) , \qquad (18)$$

where $J_T$ is the corresponding Jacobian matrix ($n \times n$), and the time complexity of computing the log determinant is $O(n^3)$.

With continuous normalizing flows the complexity can be reduces to $O(DH + D^2)$, as the change in log probabilities can be written as (Chen et al., 2018; Grathwohl et al., 2019)

$$log(p(z_{t_1})) = log(p(z_{t_0})) - \int_{t_0}^{t_1} Tr\{J_T(z_t)\}dt , \qquad (19)$$

Where the time complexity of obtaining the exact Trace is $O(n^2)$ (Papamakarios et al., 2021).

Instead of exact Trace computation, we further reduce the time complexity to $O(DH + D)$ using *Hutchinson's trace estimator* to obtain an estimate of the trace (Chen et al., 2018; Grathwohl et al., 2019; Papamakarios et al., 2021).

$$Tr\{J_T(z_t)\} = \mathbb{E}_v[v^T J_T(z_t)v] \qquad (20)$$

where $v$ is a random vector with zero mean and unit covariance. We draw binary values for $v$ from a Rademacher Distribution. Reducing the time complexity to $O(n)$ (Grathwohl et al., 2019).

However the use of numerical ODE solvers introduces an additional cost as the transformation needs to be performed $L$ times, leading to an overall time complexity of $O(L(DH + D))$. The impact of $L$ is significant in TIF, especially as the discontinuities of the derivative at the intervention points increase the number of steps taken by the numeric solver. Additionally, for the log likelihood sample, the model needs to be run forwards and backwards in time, i.e., $O(2L(DH + D))$. Finally, to

obtain the Monte Carlo estimate (Appendix D), several samples are needed, which can, however, be computed in parallel.

Finally, acknowledging that likelihood computation could become computationally heavy, we opted not to use negative log-likelihood as our loss function.

