# OpenReview forum: "Twinned Interventional Flows"
_ICLR.cc/2024/Conference — Submitted to ICLR 2024_

### Official Review · Reviewer_Tyhh · 2023-11-01

**Soundness:** 2 fair
**Presentation:** 1 poor
**Contribution:** 2 fair
**Rating:** 3
**Confidence:** 3

**Summary:**

This paper proposes a new method called twinned interventional flows (TIFs) for modeling complex dynamical systems with irregularly sampled partially missing observations and actions (interventions). To achieve this they modify continuous normalizing flows to augment them with history of latent variables that bridge the gap of partial observability and causal insufficiency (presence of hidden confounders). As a result, the model can allow counterfactual queries. Additionally, they introduce a proof that shows that training with observational and interventional data can improve performance as compared to training only on interventional data.

**Strengths:**

## Originality
- The setting (learning to model temporal potential outcomes in continuous space with partially missing observations) seems under-explored.

## Significance
- This work seems to be introducing some benefits on the tested experiments.
- It suggests a novel penalty that improves in the stiff ODE case.
- The work solves an important problem (learning to model temporal potential outcomes in continuous space with partially missing observations)

**Weaknesses:**

Weaknesses
- It’s poorly written and lacks coherence. The introduction doesn’t motivate the problem sufficiently - what is the precise setting? This should be stated in the first paragraph and not scattered around two pages of introduction (leaving a feeling of lack of focus). For example paragraph three is motivating a setting but doesn’t explicitly express it and then paragraph four start talking about something different (stiff ODE) without ever defining it or commenting on the relevance of stiff ODEs with this setting. After reading the paper several times, still cannot understand why this specific setting suffers from stiff ODEs.
- The paper combines too many ideas and it doesn’t spend much time to explain and study each of the ideas properly. For example, the penalty for mitigating stiffness is not ablated sufficiently (stiff ODEs are not even defined on the paper which makes it very hard to read)
- Related work and background section is out of focus and missing key parts as well. For example, why is causal discovery discussed? How is causal inference relevant? Is it a causality paper, if so where is the background section introducing key concepts?
- The paper is appendix heavy. A lot of the content should have been surfaced in the main text. For example, assumptions C.{1,2,3} are essential and should be surfaced on the main. Significance of theoretical results should be discussed in main as well.

**Questions:**

- What are stiff ODEs and how are they related to this setting? is it a common problem in the setting you are studying? can you show visual examples in the main text? motivate and demonstrate the issue and then solve it, that way the reader can understand why things are happening.
- What is the setting precisely? Please write a sentence explaining it as simple as possible. This shouldn't be guessed by the reader.
- in paragraph 3, "which can be counterproductive" - it's not counterproductive only, it casts the problem non-identifiable. Can you please update the text?
- What are stiff regions? This is a technical term and needs definition before used.
- Unobserved confounders is an assumption and is not explicitly stated. Please put a list of the assumptions you are making (also this is known as causal sufficiency - i'd suggest to use terms known in the community if your audience is causality)
- In related work, missing discussion on POMDPs for continuous time (e.g. "POMDPs in Continuous Time and Discrete Spaces" or "Flow-based Recurrent Belief State Learning for POMDPs"), can you please add or discuss relevance?
- In related work, causal discovery discussion is unrelated, can you please discuss the relevance to your work or remove?
- Also, are you focusing mostly on learning from offline data? If so, would discussion of offline RL be of relevance here?
- In POMDP formulation you say that rewards are not required, however if you remove rewards from the POMDP then it's just a graphical model (Decision Process requires rewards). What does the POMDP formulation buy you?
- What is the "twinned space"?
- In 3.2 you stalk about the stability of Monte Carlo estimate. Where are you using MC?
- You cite Manski 1989, can you please comment on the relevance of this paper to you work? It doesn't seem to comment on the use of "observational data helping learning the dynamics of discrete interventional settings".
- Figure 6 - The plot doesn't match the performance of the baselines in the paper. Did you use the same code? Also, how many seeds did you run? Are you plotting the standard error? I don't see any variance on your method - is this because it's stable between seeds?
- What is the take-away message from the half cheetah example? How is this experiment related to your setting?

---

> ### Author Response · Authors · 2023-11-19
>
> Thank you for your feedback. We address your questions and concerns below.
>
>
> 1. **Coherence and motivation behind the combination of ideas.**
>
>     We appreciate your feedback and understand the importance of clarity in our writing. Our objective is to address a combination of real-world problems, each of which is carefully chosen based on its relevance to applications such as medical decision-making. The selection of ideas is motivated by the need to integrate knowledge from different fields to tackle complex systems effectively.
>
>     The issue of stiffness, in particular, arose during the development of our method. Despite searching existing literature, we found no satisfactory solutions. Consequently, we devised our own approach, which demonstrated significant improvements in our experiments. Please see the visualization in Section 4.1, if it's sufficient to explain the problem.
>
>     We acknowledge the importance of conveying these motivations more explicitly, and we will make adjustments to the text to provide a clearer explanation of why these specific problems are included in the paper. If you have additional suggestions or specific areas you would like us to address, please let us know.
>
>
> 2. **Regarding stiffness in ODEs.**
>
>     There is no clear measure of stiffness in the literature and our definition, "Stiff ODEs are numerically difficult to invert," is derived from Ernst Hairer’s observation that "stiff equations are problems for which explicit methods do not work" [1], combined with insights from [2] that errors can significantly magnify when ODEs are inverted.
>
>     To address your concern, we will refine our wording and explicitly reference [1] to provide a clearer foundation for our definition. If you have any further suggestions or specific areas you would like us to enhance, please let us know.
>
>     Regarding the visualization of stiffness, it is presented in Figure 4f compared to the performance when the penalty term is employed (4e).
>
>     How would you like us to further visualize these problems?
>
> 3. **Focus of related work and background section.**
>
>     In our paper, we approach causality from a specific angle, choosing not to delve deeply into causal discovery. We mention it mainly due to its relevance to the intersection of reinforcement learning (RL) and causal fields. Instead, our primary focus lies in recognizing the causal properties of the system, particularly addressing challenges like unobserved confounders. We achieve this by incorporating these considerations into the model design, ensuring flexibility to tackle such complexities.
>
>     To accomplish our goals, we amalgamate tools from both the causal inference and RL domains. As a result, our background section delves into these topics, expecting a foundational understanding from the readers. We acknowledge the absence of exhaustive technical background knowledge in the paper, given its assumption of familiarity with these fields.
>
>     Nevertheless, we are open to adjusting our focus if you deem any particular aspect irrelevant or have specific topics in mind that should receive more attention. Your insights are valuable in shaping the paper to meet the expectations of our audience.
>
> 4. **The balance between appendixes and main body.**
>
>    We share this concern with you, however the strict page limit forces us to make some compromises.
>
>     If there are specific sections in the appendix that you believe could be integrated into the main body or should be discarded, please let us know. We are open to reorganizing the content to enhance the overall flow while acknowledging to the page limit.
>
> 5. **The setting summarized in a sentence.**
>     Our primary objective is to develop a predictive model that accurately captures the system
> dynamics under various interventions and possible (unobserved) confounders. This model should be versatile enough to estimate the likelihood of events and make use of both observational and interventional data. We believe this framework is applicable to various real-world scenarios, particularly in fields like medicine.
>
> 6. **Wording and additions.**
> Thank you for your detailed suggestions on the wording, (e.g. wording in paragraph 3, and adding the term causal sufficiency). We appreciate your feedback and will incorporate the necessary changes in the upcoming version, which we aim to upload shortly. Your understanding of the delay is much appreciated. We look forward to continuing the discussion.
>
>
> 7. **Definition of stiff regions.**
>
>     We will add this definition to the text: By stiff regions we mean the regions of weight space where the corresponding NODE becomes stiff.
>
> The response will be continued in the following message.

---

> > ### Author Response · Authors · 2023-11-19
> >
> > ## Questions and answers
> >
> > 8. *In related work, missing discussion on POMDPs for continuous time...?*
> >
> >     Thank you for highlighting these relevant references, "POMDPs in Continuous Time and Discrete Spaces" and "Flow-based Recurrent Belief State Learning for POMDPs." We appreciate your suggestion, and we will incorporate references to these papers in the section discussing RL approaches in a continuous setting.
> >
> > 9. *Also, are you focusing mostly on learning from offline data? ..?*
> >
> >     We appreciate your question. Our primary focus is on integrating additional offline data into the context of online learning. This essentially combines aspects of both offline and online learning from the RL field. Given the constraints of page limitations, we have chosen not to delve extensively into these topics, as our primary contribution lies in the theoretical results, specifically tailored for TIF and normalizing flows. If you have further suggestions on how to address these aspects more effectively within our limitations, we would be open to considering them.
> >
> > 10. *In POMDP formulation you say that rewards are not required...?*
> >
> >     Thank you for your observation. We will modify the statement to clarify that we do not explicitly discuss rewards in the POMDP formulation. This adjustment aligns more accurately with our intent. In our RL setup, TIF can serve as a predictor of the state, and our focus is on demonstrating its performance without explicit consideration of rewards. The role of rewards, would be more relevant to policy optimization, which is not the topic of this paper.
> >
> > 11. *What is the "twinned space”?*
> >
> >     The term "twinned space" refers to the combined space of observed and latent variables. We acknowledge that this concept wasn't as clearly articulated in the text, even though it is a central element of our methodology. To address this, we will incorporate a formal definition of the "twinned space" in the revised version for better clarity.
> >
> > 12. *In 3.2 you talk about the stability of Monte Carlo estimate. Where are you using MC?*
> >
> >     Thank you for bringing this to our attention. The Monte Carlo estimation is employed in the log-likelihood estimation algorithm detailed in Appendix D. In the revised version, we will explicitly reference Appendix D when discussing the use of Monte Carlo estimation and its stability.
> >
> >
> > 13. *You cite Manski 1989, can you please comment on the relevance of this paper to you work? It doesn't seem to comment on the use of "observational data helping learning the dynamics of discrete interventional settings”*
> >
> >     The citation of Manski (1989) is included in our work due to his significant contribution to the proof of bounds with binary variables. Following the example set by [3], we aimed to acknowledge the foundational work on this problem. While Manski's paper may not directly address the use of observational data in learning the dynamics of discrete interventional settings, its inclusion is more contextualized in the broader history and foundations of the problem we are addressing.
> >
> >
> > 14. *Figure 6 - The plot doesn't match the performance of the baselines in the paper. Did you use the same code? Also, how many seeds did you run? Are you plotting the standard error? I don't see any variance on your method - is this because it's stable between seeds?*
> >
> >      Yes, we utilized the same code for the experiments; however, regrettably, we encountered challenges in reproducing the results of the baselines. ODE solver errors with their method posed significant obstacles. The plot represents outcomes from five distinct random seeds, and the standard error is illustrated over these seeds. While the variance of our method is depicted with red opacity fill, covering a broad range of results, it might not be clearly visible due to its extensive coverage.
> >
> > 15. *What is the take-away message from the half cheetah example? How is this experiment related to your setting?*
> >
> >     The reinforcement learning (RL) benchmark underscores the applicability of TIF in this domain. While it may not surpass other baselines in performance, it introduces distinct advantages, such as log likelihood estimation and anomaly detection. The crucial takeaway is that TIF integrates effectively into the RL framework, demonstrating its capability to handle real-world online data.
> >
> >
> >
> > Thanks again for your review. We hope your concerns have been sufficiently addressed, and would appreciate if you could kindly consider increasing your score.
> >
> > ## References
> >
> >  [1] G. Wanner and E. Hairer, Solving Ordinary Differential Equations II (Springer,
> > Berlin, 1996), Vol. 375.
> >
> > [2] Kim, S., Ji, W., Deng, S., Ma, Y. \& Rackauckas, C. (2021), ‘Stiff neural
> > ordinary differential equations’, Chaos: An Interdisciplinary Journal of
> > Nonlinear Science 31(9), 093122.
> >
> > [3] Maximilian Ilse, Patrick Forre, Max Welling, and Joris M. Mooij. Combining interventional and ´
> > observational data using causal reductions. arXiv, 2021.

---

> > > ### Author Response · Authors · 2023-11-22
> > >
> > > Dear reviewer,
> > > we appreciate your valuable feedback, which has aided us in clarifying crucial aspects of our work. We trust that our response has effectively resolved your concerns and will hopefully contribute to an improved score. Should you have any additional questions before the discussion window concludes tomorrow, we are open to further discussion.
> > > Thank you for your consideration!

---

> > > > ### Comment · Reviewer_Tyhh · 2023-11-23
> > > >
> > > > Thank you for taking the time to address many of my questions and concerns - I appreciate your effort. However, unfortunately I still have concerns and I'd like to keep my score. Details below.
> > > >
> > > > If rewards are not of a concern or used for policy optimization, then POMDP might not the right abstraction. What's left is just a graphical model and i'm not saying this to devalue their importance, rather i'd like to emphasize that this should simplify your narrative as you could avoid the connection with RL which I'm still not convinced that it's the right way to describe your work - it would make sense to connect with RL if you care about the impact of the confounder in identifying the optimal policy. To be more clear, I think this is an important problem but I believe it would be more clear if the narrative structure was like the following:
> > > > - Here's a graphical model describing our setting
> > > > - We assume randomization as interventions (conencting with Rubin's potential outcomes framework rather than SCMs)
> > > > - There are some latent / hidden confounders and we take care using our novelty.
> > > >
> > > > Regarding what should be put in the main vs appendix, these are some suggestions:
> > > > - All of the assumptions should be stated clearly in the main and ideally in the intro.
> > > > - Algorithm is of the main interest of the reader, shouldn't be left for the appendix (assume i didn't read the appendix, would the paper make sense without it?)
> > > >
> > > > Standard error in figure 6 is quite high and overlapping between the baselines thus it's difficult to make any reliable assessments about the additional benefit of TIF over VAE-RNNs (or even simple RNNs) (acknowledged by the authors already). Also you say, "The crucial takeaway is that TIF integrates effectively into the RL framework, demonstrating its capability to handle real-world online data." - I'd avoid calling half-cheetah a real-world problem. Perhaps my hint here is that this experiment is more of a distraction than a meaningful experiment and you should save this space for adding more relevant things from the appendix.
> > > >
> > > > Also regarding the focus of related work. I still believe that RL and causal discovery is irrelevant in the sense that you don't do causal discovery and you should constraint the related work in the intersection of RL and treatment effects, which has plenty of interesting work to cite.

---

> > > > > ### Author Response · Authors · 2023-11-23
> > > > > **Thank you for your constructive feedback - please see our response**
> > > > >
> > > > > Many thanks for sharing your perspective on improving the presentation of the work. We greatly appreciate and would take these comments into account in the final version (we are not able to reflect these changes immediately since the discussion window ends very shortly). Re the points you raised:
> > > > >
> > > > > (1) Our formulation falls in the area of causal reinforcement learning, and we feel for reasons of transparency (and to reflect the contributions of the RL community in this context), it's important to emphasize the connection with POMDP.  For example, combining offline/historical/off-policy data with online policy/interventional data is very much a part of core RL, which we show the benefits of in our setting.
> > > > >
> > > > > (2) Please note that we do capture the effect of unknown confounders using latent variables,  but do not think having explicit rewards is central to the POMDP formulation here. Please note that well-studied problems such as inverse RL also do not assume availability of a good reward function. We view not having to learn the rewards as a positive aspect (simplifies the complexity of learning and training) not as negative.
> > > > >
> > > > > (3) About the RL experiment: Our objective here was to illustrate the versatility of the problem - even without any significant parameter tuning of our method, we are inarguably competitive with some strong baselines.  More importantly, none of these other methods can enable exact log-likelihood estimation and anomaly detection in RL settings as TIF can. In fact, we are not aware of any prior work in the causal reinforcement learning literature that can achieve this.
> > > > >
> > > > > We are grateful for your thoughtful comments. We will consider moving the RL experiment to Supplementary section and make space for algorithms in the main text. We will also incorporate many of your other suggestions to reorganise parts of this work.
> > > > >
> > > > > Thank you once again, and hope we adequately clarified your concerns. We would greatly appreciate and hope for stronger support from you during the AC-reviewer discussion phase.

---

### Official Review · Reviewer_dXx9 · 2023-11-01

**Soundness:** 2 fair
**Presentation:** 1 poor
**Contribution:** 1 poor
**Rating:** 3
**Confidence:** 3

**Summary:**

The paper introduces a new time-varying generative model based on conditional continuous normalizing flows, namely, twinned interventional flows (TIF). TIF operates under the assumption of a partially observed Markov decision process (POMDP) with continuous time, can be fit with both observational and interventional data, and can handle irregular and missing data. The model promises to perform predictions in the presence of unobserved confounding, to detect anomalies, to do counterfactual inference, and to facilitate reinforcement learning and treatment effect prediction. To verify the claimed properties of the model, the authors provided experiments with different benchmarks.

**Strengths:**

The paper develops a new flexible generative model for continuous-time irregularly-sampled data. It has many potential applications, as it offers a tractable log-probability at each time step and does not suffer from the limitations of fixed-dimensional embeddings. The paper also provides several important theoretical results, like the marginalisation of the infinitesimal change of variables theorem (3.1).

**Weaknesses:**

The main weakness of the paper, in my opinion, is that TIF aim at many different applications, but the lack of rigour and experimental evidence makes it unclear what this method is actually useful for.

**Lack of rigour**. The paper positions itself as both a reinforcement learning and a (causal) time-varying treatment effect estimation method but makes little effort to explain the connections between both. It inherits the assumptions, typical for reinforcement learning, i.e., POMDP, but uses it for causal benchmarks, like interventional or counterfactual predictions. For example, benchmarks for interventional predictions (like the tumor generator) are based on a different set of assumptions, e.g., three assumptions from Seedat et al. (2022), and, thus, are not suitable for TIF. Importantly, the straight application of the method to interventional and counterfactual prediction tasks raises many identifiability questions. For example, is there a need to adjust for time-varying observed confounders or to perform propensity weighting to obtain unbiased interventional predictions? Also, what is the purpose of the data with the hidden confounding (I understand, that it does not harm performance, given infinite data, but I don’t see how it could facilitate interventional predictions). Regarding counterfactual predictions, they would require even stronger identifiability assumptions [1], such as additive latent noise. Those assumptions are not properly discussed in the paper. The paper also claims to handle unobserved confounding, but in this case, without further assumptions, the causal effects are non-identifiable.

**Lack of experimental evidence**. Some key statements in the paper were not supported by the experimental evidence, for example, the claim that TIF are better than methods with low-dimensional embeddings. Also, the authors claimed that the method provides accurate density estimates, but no comparison with other density estimators was provided. The same applies to the counterfactual benchmark, where no fair comparison with other existing baselines, e.g., [2], were provided, or anomaly detection benchmark. Regarding, the RL benchmark, it does not seem like TIF significantly outperformed VAE-RNN (see Fig. 6). Also, there are no implementation details provided for the benchmarks. Additionally, I found it unfair, that the authors did not provide details on the hyper-parameter tuning and did not provide the source code of the method. E.g., it is unclear, how to choose the latent dimensionality ($n - m$) or number of MC samples. For the same reason, I cannot verify, whether the comparison between TIF and TE-CDEs was fair for the interventional prediction benchmark.

I also have several minor remarks, which are important for clarity and understanding of the paper:
- Table 1 mixes up completely different methods, aka “compares apples with oranges”, e.g., causal inference methods with reinforcement learning methods; time-varying methods and cross-sectional methods, etc. Also, it is unclear, what property “making prediction in the presence of unobserved confounders” means, as in this case, we would need to assume some sensitivity model and perform partial identification of causal effects. Additionally, there seem to be wrong entries, like the cross for De Brouwer et al. (2022) at “logp” column (this paper assumes a likelihood model, thus we can infer log-probability), or Seedat et al. (2022) at “conf” (TE-CDEs are not suitable for hidden confounding).
- Some of the notation and definitions in the paper were not properly introduced. For example, a POMDP was never formally defined. Also, what is $\hat{Q}_{t_j, t}$ in Eq. 1? What is the definition of the “privileged data”? What are “sub-flows”?
- Some of the causal inference terminology is used inconsistently. For example, counterfactual prediction is sometimes used in the meaning of the interventional (Sec. 4.2), and “counterfactual trajectories” are used instead of “interventional trajectories” (Sec. 4.1 “Counterfactual prediction”). Notably, the terms “interventional” and “counterfactual” denote fundamentally different concepts of causal inference.
- I found the usage of the terms “observational” and “interventional” data a bit confusing in this paper. Usually, the term “observational data” is used for the data, where treatment assignment depends on other observed $o$ and unobserved “l” confounders, whereas “interventional data” means a randomised control trial, i.e., no arrows leading to variables $a$ in Fig. 1. This paper, on the other hand, uses the term “observational” for the data with unobserved confounders and “interventional” for the data with all confounders observed.
- Seems like the authors confused the causal Markov condition and Markov property of POMDP, which are two different things.

references:
[1] Nasr-Esfahany, Arash, Mohammad Alizadeh, and Devavrat Shah. "Counterfactual identifiability of bijective causal models." International Conference on Machine Learning. PMLR, 2023.
[2] Hızlı, Çağlar, et al. "Causal Modeling of Policy Interventions From Treatment-Outcome Sequences." International Conference on Machine Learning. PMLR, 2023.

**Questions:**

- What is exactly meant by the ability to perform “online prediction”?
- Why is the concatsquash block important for the architecture of TIF? Couldn’t it be done simpler?

---

> ### Author Response · Authors · 2023-11-18
>
> Many thanks for your thoughtful  feedback. We  address all your comments and suggestions below:
>
> 1. **Causal reinforcement learning: connection between RL and treatment effect estimation.**
> Thank you for your comment. The connection between causal inference and RL fields is well studied by Bareinboim [1]. In our setting, the reinforcement learning set up (POMDP) is closely tied with causal treatment effect prediction; and our work can be seen as addressing two of the six main tasks in causal reinforcement learning, namely, (a) Generalized policy learning and (b) Counterfactual decision-making. Additionally, deciding when and where to intervene could be addressed with a good predictor that is continuous in nature.
>
>     Specifically, each action $a_t$ serves the role of a treatment, and time-varying confounders can be included in $l_t$. Effectively predicting treatment effects implies that the problem can be cast as an RL task, where the objective is to optimize the treatment based on the predictions. This perspective highlights the interplay between causal treatment effect prediction and RL, where a good predictor with continuous properties can inform decision-making in RL. Our POMDP formulation serves this purpose.
>
>     Hopefully this clarified the connections between these topics and we will make the according changes to the paper to highlight these topics.
>
>
> 2. **Identifiability and our  assumptions.**
> We understand your concerns about identifiability. Indeed, identifying the number of confounders or recovering a full causal graph in general is tedious and often infeasible. We also acknowledge that there are scenarios where data limitations may hinder learning of the true causal structure due to identifiability issues.
>
>     Our primary goal here is to develop a predictive model that accurately captures the system dynamics under various interventions and possible (unobserved) confounders in our setting that treats the size of the latent space as a tunable parameter, and abstracts many practical situations with confounders. We therefore appeal to an approach that does not constrain the causal hierarchy within observed and unobserved variables and can be described in terms of a continuous POMDP.
>
>     Importantly, if desired, identifiability can be achieved with appropriate additional assumptions in our setting. Indeed, one way is identification via (weighted) time-independent sampling by adapting Tennenholtz et al (2020) [2]. This, however, is susceptible to the curse of dimensionality. Instead, one can invoke a (more amenable) reduction to proximal causal inference as illuminated by Bennett and Kallus [3].
>
>     The set of assumptions provided in the paper are also consequence of this approach. The Overlap assumption by Seedat et al. [4] is listed in our set of assumption in Appendix C, as it necessary for deriving our theoretical results. However the two other assumptions by [4] are more related to the identifiability of the problem, which is not our focus here, and so were not listed. However we are happy to add these assumptions with a discussion on identifiability based on your feedback.
>
> 3. **Regarding the experimental evidence.**
>  Our primary focus in the experimental section was to showcase the various  capabilities of the proposed methood TIF, but not to imply or assert its  superiority over low-dimensional embeddings like TE-CDE [4]. We emphasize the flexibility afforded by TIF due to its many strengths inherited from continuous normalizing flows, particularly in density estimation. In this context, it is crucial to note that while density estimation is primarily a theoretical framework, the stability of the Monte Carlo estimation algorithm, as outlined in Appendix D, requires further validation. Consequently, we refrained from making comparisons with, for example, Bayesian methods.
>
>     We are grateful to you for bringing the two new papers [5], [6] to our attention, which are important and highly relevant to our topic. Unfortunately these works were not published when we worked with the background study of TIF. We are happy the see that the field is evolving and we will cite and position these papers appropriately in the final version of the paper. However, due to time constraints, benchmarking against them is unfortunately not feasible during the response period.
>
>      We believe that this work already makes significant contributions, and lays the foundation to investigate aspects such as those pointed by you as future work. Based on your feedback, we will also be sure to underscore the motivation underlying each of our experiments.
>
> The response will be continued in the following message.

---

> > ### Author Response · Authors · 2023-11-18
> >
> > 4. **Hyperparameter tuning.**
> > Thank you for pointing out the missing details on hyperparameter tuning (owing to space constraints). We appreciate your understanding, and we will certainly include this information in the upcoming revision shortly. In the meantime, we provide below a brief overview of our hyperparameter tuning process.
> >
> >     We conducted all our experiments with extreme care and diligence to be fair to the baselines. We would be happy to share our code upon request and release it upon the acceptance of this work.
> >
> >     For the pendulum data, hyperparameters such as the latent dimension were fine-tuned, and the same configuration was applied to the tumor growth experiment. We note that more extensive hyperparameter tuning could potentially lead to further improvements for TIF. The learning rate was determined through multiple runs with a small number of epochs.
> >
> >     In the case of the RL benchmark, the latent dimensionality and model size were aligned with those of other benchmarks, with the latent dimension being an educated guess based on the dimensions of the Half Cheetah task. The primary parameter tuned in this scenario was the learning rate, with approximately five different options tested for each setup.
> >
> >     We hope this brief overview provides clarity, and we will ensure to include a more detailed account in the revised version of the paper. If you have any additional suggestions or queries, we are happy to discuss further.
> >
> >
> > 5. **Regarding Table 1.**
> > We position ourselves at the intersection of different fields, and specifically in the realm of causal reinforcement learning. Consequently, the topics covered in Table 1 are deliberated across a diverse range of models, as you rightly pointed out. Despite these discussions traditionally occurring within distinct fields, we view the problems in Table 1 as pertinent to real-world issues, arising e.g., in medical settings, that ought to be addressed in a more holistic way than the conventional paradigm. Therefore, in our perspective, there is merit in addressing them collectively, which the previous literature has not done as illustrated in Table 1.
> >
> >     With the property “making prediction in the presence of unobserved confounders” we refer to whether or not the possible confounders are taken into account in the model design and discussed in the paper.
> >
> >     Regarding what you suspect as potentially incorrect entries, please allow us to elucidate our reasoning.   De Brower et al. use decoder encoder decomposition and are therefore limited to obtaining variational bounds of the log-likelihood, while TIF extends continuous normalizing flows and therefore can obtain the exact log-likelihood estimate.
> >
> >     Seedat et al. do make predictions in the presence of confounders as they say "TE-CDE uses domain adversarial training to learn a representation that adjusts for time-depending confounding and hence is suitable for causal estimation". However, the column under consideration does not stand for causal discovery, but rather whether the presence of possible confounders is discussed and taken into account in the design of the method.
> >
> >     Thanks for your comments on this. We will carefully revisit the wording of the table description to avoid any misunderstandings.
> >
> >
> > 6. **Notation and definitions.**
> > We apologise for any confusion due to definitions and choice of our notation. We would be happy to accommodate any suggestions you might have to explicate the POMDP framework beyond the current mathematical description of section 2.1.
> >
> >     We will also clarify the definition of $\hat{Q}_{t_j,t}$, to highlight that it refers to the model approximation of the change in the model state. It is a necessary term as it relates to these "sub flows". As the observations and actions are observed at discrete time points, we need to account for discontinuities in the derivative of the flow at these points. Thus the flow needs to be divided into continuous sub flows that must be summed over to obtain the approximation of the state at time t. We will be sure to include this at the beginning of section 3.
> >
> >     The term "privileged data" has been defined in Section 2.1 as "observations of the state that are not available for the intervening agent," and it is visually represented in Figure 1 with the red arrow. To provide further clarity, we can include a brief reminder or illustration in subsequent sections to ensure consistent understanding. How would you prefer us to elaborate on this matter?
> >
> > The response will be continued in the following message.

---

> > > ### Author Response · Authors · 2023-11-18
> > >
> > > 7. **Use of causal inference terminology.**
> > > We appreciate your suggestion. In response, we will modify the wording in Section 4.2 to use "interventional trajectories" as it aligns more accurately with the setup. It's worth noting that while our terminology choice may differ from Seedat et al [4], we believe this adjustment will enhance clarity of our presentation.
> > >
> > >     We argue that in section 4.1 (Counterfactual prediction) the term counterfactual trajectory is used correctly. Please refer to supplementary materials H for an in-depth discussion on this matter. The meaning of counterfactual trajectory is also illustrated in Figure 2 b. While we are fully aware of the distinction between the terms interventional and counterfactual, we welcome any specific suggestions or preferences you might have to enhance the presentation.
> > >
> > > 8. **Terms “observational” and “interventional” data**:
> > > We are sorry for any misunderstanding. We indeed use wording observational to refer to data where the treatment assignment depends on the observed and confounding variables and interventional on the case where the information about these confounders is missing (please see  2.1. where these terms are defined). We do follow the common standard as you described it, but instead of randomized policy in interventional data we state that the policy can be arbitrary and can also be learned via reinforcement learning.
> > >
> > >     While we recognize that "interventional" data often refers to randomized trials, we have found it challenging to identify a more suitable term for our setting. We are, however, open to any suggestions you might have in this regard.
> > >
> > > 9. **Causal Markov condition and Markov property of POMDP**:
> > > We would like to respond to ”authors confused the causal Markov condition and Markov property of POMDP", emphasising that our usage is correct.
> > >
> > >     By definition, POMDP satisfies the Markovian assumption with respect to $s_t$ [5]. Given our assumption that $o_{t_j} ,l_{t_j}$ are sufficient to fully describe $s_{t_j}$, the Markovian assumption also holds with respect to $o_{t_j} ,l_{t_j}$.    The Markovian causal model is illustrated in Figure 3.3 and it satisfies the parental Markov condition.
> > >
> > >     Thus, both of these terms can be used in this context with parental Markovian condition arguably being more suitable for the causal discussion. Additionally, the most important conditional independences are outlined in the appendices for clarity. The matter is also discussed in Appendix C.
> > >
> > >
> > > 10. **Meaning of “online prediction”.**
> > > We appreciate your inquiry. In our context, "online prediction" signifies that information about the future state of the system is not available during the prediction process. Consequently, methods that rely on implicit (ODE) solvers or, for example, splines are not applicable. This notion of online prediction is crucial in real-world scenarios and reinforcement learning (RL) applications where foresight into future states is unavailable.
> > >
> > > 11. **Regarding the concatsquash block**:
> > > Thank you for bringing up the ConcatSquash block. Indeed, our framework does not necessitate its use.  Its selection as a component in our architecture was based on its demonstrated performance in the related literature on normalizing flows and neural ODEs. While we believe it intuitively fits our needs, we acknowledge that alternative layouts are also feasible.  Furthermore, tuning these architectures would likely enable further empirical and computational improvements.
> > >
> > > We are grateful for your feedback that has helped us illuminate several subtle aspects of this work. We hope our response has sufficiently addressed your concerns, and would appreciate if you could kindly consider increasing your score.
> > >
> > > ## References:
> > >
> > > [1] Elias Bareinboim (2020), Causal reinforcement learning. Conference Tutorial
> > > at 37th International Conference on Machine Learning, 2020, 13-18
> > > July,
> > > URL: https: // crl. causalai. net/
> > >
> > > [2] G. Tennenholtz, S. Mannor, and U. Shalit. Off-policy evaluation in partially observable environments. In Proceedings of the 33rd AAAI Conference on Artificial Intelligence (AAAI), 2020.
> > >
> > > [3] Andrew Bennett \& Nathan Kallus  (2021), ‘Proximal reinforcement learning: Efficient off-policy evaluation in partially observed markov decision processes’,
> > > arXiv: 2110.15332.
> > >
> > > [4] Nabeel Seedat, Fergus Imrie, Alexis Bellot, Zhaozhi Qian, and Mihaela van der Schaar. Continuous time modeling of counterfactual outcomes using neural controlled differential equations. In
> > > the 39th International Conference on Machine Learning, ICML 2022, 17-23 July,
> > > Baltimore, USA, volume 162, pp.19497–19521.
> > > PMLR, 2022.
> > >
> > > [5] Nasr-Esfahany, Arash, Mohammad Alizadeh, and Devavrat Shah. "Counterfactual identifiability of bijective causal models." International Conference on Machine Learning. PMLR, 2023.
> > >
> > > [6] Çağlar Hızlı et al. "Causal Modeling of Policy Interventions From Treatment-Outcome Sequences." International Conference on Machine Learning. PMLR, 2023.

---

> ### Author Response · Authors · 2023-11-22
>
> Dear reviewer, we are grateful for your constructive feedback that helped us elucidate some important aspects of this work. We hope our response has sufficiently addressed your concerns, and hope it translates into an increased score. We are also happy to discuss further if you have any further questions before the discussion window ends tomorrow. Many thanks!

---

> > ### Comment · Reviewer_dXx9 · 2023-11-23
> >
> > Dear authors, I appreciate your effort in trying to address my comments. Yet, there are still many unresolved concerns.
> >
> > '1'. Thank you for providing a reference to [1] Elias Bareinboim (2020). Still, I think that the Related Work section lacks structure and the contrast between what is novel in the paper vs what was already in the literature. Also, it is still unclear to me, what is the main contribution of the paper wrt. previous works. For example, the authors claimed
> > > We focus on motivating the benefits of flow models for POMDPs.
> >
> >    Does this imply that the approach of the paper is simply to select the RL tasks, where normalizing flows perform well? If so, this paper needs to provide a rigorous comparison with **all the existing baselines** for **each** specific RL task (or at least one).
> >
> > '2'. What exactly do you mean here:
> > > However the two other assumptions by [4] are more related to the identifiability of the problem, which is not our focus here, and so were not listed.
> >
> >    How is the identifiability of the problem not important for this paper?
> >
> > '3'. I acknowledge, that the provided by me examples are very recent and according to the ICLR policy are allowed not to be mentioned in the current paper. Nevertheless, [6] Çağlar Hızlı et al. provided a related works table (Table 3), where there are methods, published before May 2023. For example, "Noorbakhsh, K. and Rodriguez, M. G. Counterfactual temporal point processes. In Advances in Neural Information Processing Systems, volume 35, 2022." and  "Hua, W., Mei, H., Zohar, S., Giral, M., and Xu, Y. Personalized dynamic treatment regimes in continuous time: A Bayesian approach for optimizing clinical decisions with timing. Bayesian Analysis, 1(1):1–30, 2021." It seems to me that those methods are relevant to the current paper and have to be properly discussed in the related work and included in the experiments.
> >
> > '4'. Thank you for the brief explanation, but I would still like to know the exact tuning ranges and tuning criteria. This is very important, as I cannot verify the transparency and reproducibility of the results.
> >
> > '5'. To the best of my knowledge, TE-CDE considers all the confounders to be observed (see Assumption 3 (Continuous-time sequential randomization)).
> >
> > '9'. Under the causal Markov condition, I meant that there is no hidden confounding in the causal diagram (Fig. 1). And under the Markov property of POMDP, I implied that the future states are independent of the past, given the most recent state. Correct me if I am wrong, but those two assumptions are in general different. Specifically,  we can have the causal Markov condition, but dependencies between the future states and the whole history given the current state, and vice versa,  a POMDP with Markov property but with hidden confound and, thus, non-Markovian SCM.
> >
> > '11'. "Indeed, our framework does not necessitate its use." By the principle of Occam's razor, shouldn't simpler architecture be considered then? At least I would expect some form of the ablation study.
> >
> > As there are still many unresolved issues, I tend to keep my score the same.

---

> ### Author Response · Authors · 2023-11-23
>
> 1.  Thank you - we'll structure the related section better based on your comments.
>     Our main contribution is to use flow models in an augmented space that couples observations and latent variables including unknown confounders, to enable exact (not just lower bounds on the) log-likelihood estimates as well as predict treatment effects.
>     We're not aware of any previous work in literature that can handle both these tasks simultaneously in the presence of unknown confounders. Since we do not observe the unknown variables only part of the information is visible and the state evolves continuously  over time, so the problem can be investigated in a POMDP  setting.
>     We would like to emphasize that our goal is not to cherrypick any RL tasks, but to motivate this work as a generic framework that can also be leveraged in RL settings.
>     **In our RL task, we already compare with Latent-ODE which is a strong normalizing flow-based baseline.**
> 2. Identifiability is certainly important, however, identifiability results in our POMDP setting can be derived using e.g., proximal causal inference with additional assumptions. Since such theoretical results on identifiability are already known, and these additional assumptions do not play any role otherwise in the entire setup (e.g., for the purpose of likelihood estimation), we did not elaborate on these to keep the exposition focused.  We will make this clear based on your comments.
> 3. Thank you for pointing these papers to us. We will appropriately position the contributions of these works. Please note that neither of these papers allow for exact log-likelihood estimation in the presence of unknown confounders
> 4. The fine tuning of the learning rate for the tumor effect prediction was done with values 0.01- 0.0001 for TIF and the values reported in literature were used for TE-CDE. **Based on your comment, we have now also shared the code of this experiment for reproducibility**.
> 5. To our understanding the assumption 3 by Seedat et.al. means that the current history is sufficient for making accurate counterfactual predictions without incorporating the information about the potential factual outcomes. This does not mean that there is no confounding effect in the system, but rather that the history observations are sufficient to accurately model them. But we will re visit this and make corrections if needed.
> ---
> 9.
>     Thank you for elaborating this. We see it as: there is unobserved confounding in causal diagram (see $l$ in in Figure 1 and observe that red arrow is present only in the observational data) however the markovian assumption in POMDP is satisfied in respect to the full state $s=[l,o]$, thus if we obtain accurate estimate of $l$ we can make accurate predictions based on the current state. The parental markovian condition follows from the structure of of the DAG, that is, each variable is independent on all its nondescendants, given its parents (Pearl & Verma 1995).
>     However, we will revisit our wording and make sure that there is no ambiguity and the terms are conveyed correctly.
> ---
> 11. Thank you.  Finding simpler components that could be alternatives to concatsquash block is certainly an interesting direction, but requires further research. In general, beyond Occam's razor (as often implemented e.g., by instantiating the minimum description length principle - which loosely speaking amounts to choosing a simple hypothesis from a set of candidate hypotheses that all fit the training data well), we know from learning theory foundations that there is an inherent tension between expressivity and generalization when considering different families of architectures (or class of hypotheses) and the sweet-spot lies where the two are roughly balanced.
>     Unfortunately, the generative ML community is still trying to work out the generalization bounds for flow based models, so a detailed principled analysis of the impact of different components is very much an open question that this work inherits.
>     Please do note that several works have shown the empirical merits of using concatsquash block in similar settings, i.e., for embedding in dynamic systems, so we adopted this choice to be consistent with the prevailing paradigm.
>
>
> Based on your comments, we'll be happy to include a discussion on this in the Supplementary section as well.
> Thank you again for your thoughtful comments, and we hope our response here satisfactorily addresses your concerns and translates into an improved score.

---

### Official Review · Reviewer_ELA6 · 2023-11-07

**Soundness:** 3 good
**Presentation:** 2 fair
**Contribution:** 3 good
**Rating:** 6
**Confidence:** 3

**Summary:**

This paper proposes a novel approach called Twinned Interventional Flows for estimating causal effects of interventions in continuously evolving systems, based on a modified conditional normalized flow model. The authors introduce the concept of "twinning" and use conditional continuous normalizing flows to model system dynamics, obtaining accurate density estimates. The proposed method is flexible and can be applied to various tasks, especially for counterfactual prediction while combining both experimental and observational data. The paper also presents theoretical and empirical results demonstrating the efficacy of the proposed framework.

**Strengths:**

- (Originality & Quality) This paper presents an interesting and novel concept, "twinning", which augments partial observations with additional latent variables including confounders as a state and treats interventions as actions in a POMDP setting.
    - The proposed approach is based on a modified conditional normalized flow (CNF) model with bijective mapping that incorporates the twinning of observation and unobserved latent representation. This naturally can be extended to address counterfactual inference problem.
    - This approach seems pretty versatile. Apart from addressing counterfactual inference, it also generalizes to data with missingness or irregular sampling.
    - The paper presents a solid theoretical analysis on the property of the proposed approach. This further justifies the technical soundness.

- (Significance & Potential Impact) Since the focus of this work is to generalize the conventional CNF model to the counterfactual inference setting where experimental and observational data can be use in conjunction. This work can be used in many continuous causal inference problems. The versatility in its design can potentially contribute to a broad impact.

**Weaknesses:**

- (Presentation & Clarity) Despite the technical soundness, the paper seems difficult to follow, in lack of intuitive illustration of both newly introduced concepts or adopted approaches & notations. This makes it hard for readers to follow the logical reasoning and motivation for each specific modeling proposal.
    - The method section mainly focuses on "what" -- what was designed in the proposed approach, however without enough high-level intuitive illustration of "why". Similarly, in experiment section, the purpose, tasks and characteristics of the chosen dataset were not clearly presented, which makes it difficult to understand the motivation and analysis along without checking the external references.
    - The above could be partially due to limited space. But i believe the presentation can be improved.

- (Complexity and Limitation in Scaling) Bounded by the invertibility design, the proposed normalized flow based model TIF does not abstract input data into lower-dimensional latent space with encoder. All transformation/mapping happens in the same dimensions. This would limit generalizing the model to complex data with high dimensionality (e.g, clinical data, EHR, images). Due to the high time complexity of matrix inversion (in general O(n^3)), the computation cost would be significantly heavy for either large sample size or high data dimensionality.

**Questions:**

- What's the overall time complexity of TIF?

---

> ### Author Response · Authors · 2023-11-19
>
> Many thanks for your constructive feedback. We address all your concerns and questions below:
>
> 1. **Improving presentation and clarity.**
>
>     Thank you for your inputs. We appreciate your insights, and are currently working on enhancing readability, including, by incorporating more intuitive reasoning into the methods section and elucidating the experimental section as suggested. The revised version will be uploaded shortly.
>
> 2. **Time complexity and scaling.**
>
>     Thank you for the opportunity to shed light on the computational aspects of the proposed method.  We would like to emphasize that TIF does not necessitate full matrix inversion (which would indeed have been computationally expensive with a complexity of $O(D^3)$, where $D$ denotes the data dimension), instead inheriting the more tractable computation needs of continuous normalizing flows that can be solved more efficiently by integrating backward in time [1].
>
>     Furthermore TIF inherits from the continuous flows normalizing a more efficient way of estimating the log likelihood.
>
>     If the cost of evaluating TIF is $O(DH)$ where $D$ is the dimensionality of the data, and $H$ is the largest hidden dimension in the concatsquach block, then the time complexity of one transformation using a naive normalizing flow approach would be $O(DH + D^3)$
>
>     The cubic format arises from using the change of variables to obtain complex densities $z \sim p_z(z)$ from a simple distribution $u \sim p_u(u)$ with transformation $T(u) =x$, as described in [1,2]
>
>     $log(p_z(z))=log(p_u(u)) - log(|det J_T(u)|) ~, $
>
>     where $J_T$ is the corresponding Jacobian matrix ($n \times n$), and the time complexity of computing the log determinant is $O(n^3)$.
>
>
>     With continuous normalizing flows the complexity can be reduces to $O(DH + D^2)$, as the change in log probabilities can be written as [3, 2]
>
>      $log (p(z_{t_1}))= log (p(z_{t_0})) - \int_{t_0}^{t_1} Tr \{J_{T}(z_t) \} dt~,$
>
>     Where the time complexity of obtaining the exact Trace is $O(n^2)$ [1].
>
>     Instead of exact Trace computation, we further reduce the time complexity to $O(DH + D)$ using \textit{Hutchinson’s trace estimator} to obtain an estimate of the trace [1,2,3].
>
>     $Tr \{ J_{T}(z_t) \} = E_v [ v^T  J_{T}(z_t) v ]$
>
>     where $v$ is a random vector with zero mean and unit covariance. We draw binary values for $v$ from a Rademacher Distribution. Reducing the time complexity to $O(n)$ [2].
>
>     However the use of numerical ODE solvers introduces an additional cost as the transformation needs to be performed $L$ times, leading to an overall time complexity of $O(L(DH + D))$.  The impact of $L$ is significant in TIF, especially as the discontinuities of the derivative at the intervention points increase the number of steps taken by the numeric solver. Additionally, for the log likelihood sample, the model needs to be run forwards and backwards in time, i.e., $O(2L(DH + D))$. Finally, to obtain the Monte Carlo estimate (Appendix D), several samples are needed, which can, however, be computed in parallel.
>
>
> Thank you once again for your valuable comments. We will incorporate the discussed topics into the updated version, scheduled for upload by Monday at the latest. We appreciate your engagement and are open to continuing this discussion with you.
>
>
> ## References
>
> [1] Papamakarios, G., Nalisnick, E. T., Rezende, D. J., Mohamed, S. \& Laksh minarayanan, B. (2021), ‘Normalizing flows for probabilistic modeling and
> inference.’, Journal of Machine Learning Research 22(57), 1–64.
> URL: https: // jmlr. org/ papers/ v22/ 19-1028. html
>
> [2] Will Grathwohl, Ricky T. Q. Chen, Jesse Bettencourt, Ilya Sutskever, and David Duvenaud. Ffjord:
> Free-form continuous dynamics for scalable reversible generative models. In International
> Conference on Learning Representations, ICLR 2019, 6-9 May, New Orleans, USA, Conference
> Track Proceedings, 2019. URL https://openreview.net/forum?id=rJxgknCcK7.
>
> [3] Ricky T. Q. Chen, Yulia Rubanova, Jesse Bettencourt, and David K Duvenaud. Neural
> ordinary differential equations. In Advances in Neural Information Processing Systems,
> NeurIPS 2018, 2-8 December, Montr´eal, Canada, volume 31. Curran Associates, Inc.,
> 2018. URL https://proceedings.neurips.cc/paper_files/paper/2018/
> file/69386f6bb1dfed68692a24c8686939b9-Paper.pdf

---

> > ### Author Response · Authors · 2023-11-22
> >
> > Dear reviewer, we want to express our appreciation for your thoughtful feedback, which has significantly contributed to clarifying crucial aspects of our work. We trust that our response effectively addresses your concerns and positively influences the final evaluation. If you have any additional questions before the discussion window concludes tomorrow, we would be more than happy to engage in further dialogue. Thank you for your insightful evaluation and consideration!

---

### Meta-Review · Area_Chair_FnUc · 2023-12-10

**Metareview:**

Several reviewers raised many points of criticism that address the novelty and depth of the contribution, a lack of formal rigour and unconvincing experimental evaluation. Honestly, I share most of these concerns, and I also think thatmost of them still remain after the  the rebuttal and discussion phase. Therefore I recommend rejection of this paper.

**Justification For Why Not Higher Score:**

Unclear novelty / depth of the contribution.

**Justification For Why Not Lower Score:**

N/A

---

### Decision · Program_Chairs · 2024-01-16

Reject